# Steroids and TRP Channels: A Close Relationship

**DOI:** 10.3390/ijms21113819

**Published:** 2020-05-27

**Authors:** Karina Angélica Méndez-Reséndiz, Óscar Enciso-Pablo, Ricardo González-Ramírez, Rebeca Juárez-Contreras, Tamara Rosenbaum, Sara Luz Morales-Lázaro

**Affiliations:** 1Departamento de Neurociencia Cognitiva, División Neurociencias, Instituto de Fisiología Celular, Universidad Nacional Autónoma de México, Ciudad de México 04510; Mexico; amendez@ifc.unam.mx (K.A.M.-R.); oenciso@ifc.unam.mx (O.E.-P.); rjuarez@ifc.unam.mx (R.J.-C.); trosenba@ifc.unam.mx (T.R.); 2Departamento de Biología Molecular e Histocompatibilidad, Hospital General Dr. Manuel Gea González, Secretaría de Salud, Ciudad de México 14080, Mexico; ricardo.gonzalezr@salud.gob.mx

**Keywords:** TRP channels, steroids, gene expression

## Abstract

Transient receptor potential (TRP) channels are remarkable transmembrane protein complexes that are essential for the physiology of the tissues in which they are expressed. They function as non-selective cation channels allowing for the signal transduction of several chemical, physical and thermal stimuli and modifying cell function. These channels play pivotal roles in the nervous and reproductive systems, kidney, pancreas, lung, bone, intestine, among others. TRP channels are finely modulated by different mechanisms: regulation of their function and/or by control of their expression or cellular/subcellular localization. These mechanisms are subject to being affected by several endogenously-produced compounds, some of which are of a lipidic nature such as steroids. Fascinatingly, steroids and TRP channels closely interplay to modulate several physiological events. Certain TRP channels are affected by the typical genomic long-term effects of steroids but others are also targets for non-genomic actions of some steroids that act as direct ligands of these receptors, as will be reviewed here.

## 1. Introduction

### 1.1. TRP Channels

Transient receptor potential (TRP) channels are transmembrane protein complexes of a large family of non-selective cation channels divided into seven subfamilies based on their sequence homology: TRPC (canonical), TRPV (vanilloid), TRPM (melastatin), TRPP (polycystin), TRPML (mucolipin), TRPA (ankyrin) and TRPN (no mechanoreceptor). Most of the members of these TRP subfamilies are well represented both in invertebrates and mammals, except for the TRPN channel that is exclusively found in invertebrates and fish [1], Figure 1. 

TRP channels initially were discovered as important effectors of phototransduction in the cells of the eyes of the Drosophila fly [2]. Years later, the identification of the first human homolog for this protein, the transient receptor potential “canonical 1” (TRPC1) and the isolation of the rodent TRP channel known as the “capsaicin” receptor (TRPV1) was achieved [3,4], confirming that these channels are broadly distributed in the animal kingdom and opening a wide research field for the study of these channels. 

The recently obtained high-resolution structures for several of these channels have shown that they are formed by four subunits (tetramer) [5,6], Figure 1, where each monomer is a six-pass transmembrane protein (S1–S6), with the N- and C-terminus intracellularly localized [6]. The subunits contain a re-entrant loop located between S5–S6, forming the pore or ion conduction pathway. Some of these channels contain ankyrin repeat domains (ARDs) in the N-terminus and a TRP-domain (a highly conserved 25-amino-acid α-helix structure parallel to the inner plasma membrane) at the C-terminus [5,6], Figure 2. 

In general, TRP channel structures resemble that of the four-fold symmetry that has been observed in potassium channels; however, TRP channels lack the presence of several positively-charged amino acids localized at the S4 of voltage-gated ion channels and thus they are weakly voltage dependent [3,7]. 

Extensive efforts have focused on determining the relationship between their structure and their function; additionally, the identification of several regulators of the activity of these channels has provided evidence for their wide physiological importance [8,9]. These regulators act directly or indirectly on TRP channels to modulate their function or expression. Most of these regulators are compounds of a synthetic origin but there are also some that are endogenously produced. Among the latter, steroids have been found to modulate them and affect the physiology of the cells where TRP channels are expressed. 

During the last fifteen years, several reports have shown how endogenously-produced steroids regulate the TRP channels, however, there are only a few reviews on this topic [10,11,12,13]. Since this is a rapidly progressing field, in this review we cover several new advances in this theme and we highlight the experimental evidence that has demonstrated that there is indeed a close relationship among TRP channels and some steroidal hormones as well as the effects of this on their physiology.

### 1.2. Brief Description of the Steroids that Modulate TRP Channels

Steroids are compounds that contain a fundamental structure composed of a skeleton of cyclopenta(α)phenanthrene [14], Figure 3 and that are produced from a common precursor, cholesterol, which also directly regulates the function of some TRP channels (i.e., TRPV1 channel) [15,16]. Several enzymatic steps have primordial roles within steroidogenesis, where the first product of this is pregnenolone, which is considered a prohormone since all steroids are derived from this molecule [17], Figure 3. Although it has been reported that pregnenolone modifies microtubule assembly since it directly interacts with microtubule-binding protein 2 (MAP-2) [18], no nuclear receptor for this steroid has been identified to date and, thus, genomic actions of this steroid have not been yet reported.

Interestingly, some derivates of pregnenolone are active metabolites, such as 17-hydroxipregnenolone, allopregnanolone and pregnenolone sulfate (PregS) [19]. The discovery of PregS as a metabolite exhibiting a higher concentration in the brain than in plasma [20], together with the discovery that the cholesterol side-chain cleavage enzyme is localized in the white matter of the brain [21] as well as the fact that PregS levels remain abundant in the brains of gonadectomized female rodents [20], have strengthened the concept of PregS acting as a neurosteroid (steroid synthesized in the nervous system). Furthermore, it was demonstrated that PregS is a potent antagonist of GABA (γ-aminobutyric acid) receptors showing that this compound can modify neuronal excitability [22]. In addition, it is also important to note that PregS modifies the function of other ion channels including, NMDA (N-methyl-D-aspartate), glycine and AMPA (α-amino-3-hydroxy-5-methyl-4-isoxazolepropionic acid) receptors and of some TRP channels [23,24,25,26]. 

Pregnenolone is the precursor of progesterone (P4) Figure 3, a steroid with key functions in the maintenance of pregnancy [27]. P4 is synthesized in the ovaries, adrenal gland, uterus, placenta and nervous system (hence, it is also considered a neurosteroid) [27,28]. It is essential for reproductive functions such as oocyte maturation, ovulation, implantation, proliferation and differentiation during the reproductive cycle and in the maintenance of pregnancy, as mentioned before [27]. In the menstrual cycle, the late luteal phase is characterized by displaying higher levels of P4, although during pregnancy its plasmatic concentration rises progressively with approximately 250 mg produced per day [29]. 

P4 exerts its effects primarily through the classical genomic pathway, which requires the involvement of nuclear progesterone receptors A and B (PRA and PRB). In response to ligand stimulation, the ligand-bound progesterone receptor binds to gene regulatory regions to modulate the transcription of progesterone-responsive genes [27,30]. However, progesterone also exhibits non-genomic effects that occur through non-classical transmembrane progesterone receptors and sigma receptors (Sig-R) [31,32].

Apart from P4, other steroids that play specific roles in the reproductive system are the androgens, a group of steroids important for male gonadal function [33]. These include testosterone, dihydrotestosterone (DHT), dehydroepiandrosterone (DHEA), androstenedione and androstenediol [33,34], Figure 3. Testosterone is mainly synthesized in the testis; it is the principal circulating androgen and the precursor of dihydrotestosterone, the most potent androgen [34]. In addition, the testis and the adrenal cortex produce other androgenic steroids such as androstenedione, androstenediol and DHEA [34]. Furthermore, DHEA and its sulfated form are also neurosteroids since they are synthesized in the nervous system where they modify neuronal physiology similarly to PregS [35]. 

Androgens are essential lipids for the development and maintenance of sexual characteristics and their effects are classically mediated through the androgen receptor (AR). Testosterone and DHT have a high affinity for AR, while DHEA and androstenedione function as partial agonists of this receptor. AR plays a role as a transcription factor when it is activated by its ligands and, this, in turn, regulates transcription of target genes through classical genomic effects [36,37,38]. Additionally, androgens can activate signaling pathways in a fast and non-classical fashion modifying the function of some TRP channels, as will be reviewed here [39]. 

Likewise, androgens are used for the biosynthesis of estrogens, Figure 3. These steroids maintain the female geno- and phenotypes and they also contribute to a wide variety of biological functions ranging from sexual differentiation, metabolic regulation, cardiovascular function, bone resorption to modulation of neuronal circuits that control behavioral responses [40].

Estrogen biosynthesis is catalyzed by aromatase, an enzyme that converts androstenedione and testosterone to produce estrone (E1) and 17β estradiol (E2), respectively. E2 is the most potent estrogen and shows the highest affinity for estrogen receptors, as compared to estrone [41]. 

Estrogens exert some of their genomic effects through estrogen receptors α and β (ERα and ERβ) [42]. The activation of these receptors induces complex events that culminate in the regulation of transcription of their target genes, constituting the classical long-term effect mediated by estrogens [43].

In addition, estrogens mediate non-genomic signaling pathways through another important receptor, the G protein-coupled receptor 30 (GPR30), also knowns as G-protein-coupled estrogen receptor 1 (GPER1). This receptor exerts rapid estrogen responses through a G_αs_ protein-coupled receptor that increases intracellular calcium and stimulates adenylate cyclase to produce cAMP (cyclic adenosine monophosphate), the regulator of protein kinase A (PKA) and the cAMP response element-binding (CREB) transcription factor. Interestingly, PKA activation is an important mechanism in the regulation of the function of some TRP channels [44]. 

Finally, through steroidogenesis the glucocorticoids and the mineralocorticoids are also biosynthesized [45], Figure 3. Among them is the mineralocorticoid aldosterone, which has been shown to influence the expression of some TRP channels. This steroid is highly synthesized in the glomerular zone of the adrenal cortex and is the main mineralocorticoid in influencing the appropriate reabsorption of electrolytes and water [45]. This steroid is physiologically relevant because it regulates blood pressure by producing the reabsorption of sodium and the excretion of potassium in the kidneys. Its effects are classically mediated through the mineralocorticoid receptor (a transcription factor) [46]; the interaction of aldosterone with this receptor produces changes in the expression of specific target genes (as the electrogenic Na^+^/K^+^ ATPase and some TRP channels).

The above brief description of the characteristics of steroids summarizes how and where these hydrophobic molecules are produced and also refers to the general mechanisms of action of these molecules being divided in two main ways: classical long-term genomic and non-classical short-term effects. Notably, both of them have been reported to regulate the expression and influence the function of TRP channels. 

## 2. TRPV Channels and Their Regulation by Steroids 

The TRP vanilloid subfamily contains six members (TRPV1-6), Figure 1. The name of this subgroup was coined by the first member identified, the TRPV1 channel, which is the receptor for capsaicin, a vanilloid organic compound that produces a burning sensation when we eat hot chili peppers [4]. Although the members of the TRPV subfamily show similitudes in their sequences and structures, the binding pocket for capsaicin can only be found in the sequence of the TRPV1 channel. Thus, this structural feature provides high selectivity for TRPV1 activation by this vanilloid compound [47]. 

This subfamily contains four members that are classified among thermo-TRP channels (TRPV1-4) [48], since they are activated by changes in temperature ranging from warm (i.e., TRPV4) [49,50] to hot (i.e., TRPV1-3) [4,51,52]. In addition, these non-selective cation channels are activated by chemical, mechanical and osmotic stimuli [4,51,52,53] and they are expressed in sensory neurons and many other tissues. The other two members, TRPV5-6 are highly selective calcium channels, mainly expressed in epithelial and bone cells playing important roles in the physiology of the kidney, intestine and bone [54,55]. 

The predicted structure for these channels was confirmed through single-particle cryo-electron microscopy (cryo-EM). This tool enabled the determination of the first structure of a member of the TRP family, the TRPV1 channel with a resolution of 3.27 Å [5]. This strategy also facilitated solving the structures of other members of this family of ion channels [56,57,58]. 

These structures confirmed the typical array of these channels as tetramers where the monomer is a six-pass transmembrane (S1-S6) protein with an N-terminus containing a tandem of six-ankyrins repeats, and a C-terminus comprising the TRP-box, with both of these last domains being intracellularly located, Figure 2. 

The polymodal nature of these channels, has allowed for the identification of several chemical stimuli that directly interact with specific binding-sites situated along with their sequences or that activate them through specific cell-signaling pathways. Among these are some compounds, such as steroids, that regulate TRPV expression, influencing the physiological roles of these channels. 

### 2.1. Regulation of TRPV1 Expression by Estrogens

The TRPV1 protein is highly expressed in medium- and small-diameter sensory neurons from dorsal root ganglion (DRG) and trigeminal ganglia (TG), where it is closely involved in the transduction of painful signals [4,59]. Growing evidence supports the role of TRPV1 as a modulator of sexual dimorphism in pain perception through estrogen actions. For instance, estradiol seems to be responsible for a higher prevalence of certain pain states in women, as compared to men [60]. Current evidence suggests that steroids exert at least part of these effects via TRPV1 channels [61]. A first attempt to demonstrate the relationship between TRPV1 channels and estrogens was achieved by using ERα and ERβ knockout mice-model, where these transgenic animals displayed downregulation of TRPV1 protein levels [62]. This led to other studies where the effects of estradiol on pain mediated through TRPV1′s activation were investigated. 

Specifically, behavioral assays clearly showed that female rats are more sensitive to pain responses produced by capsaicin than their male counterparts. Moreover, ovariectomized rats exhibited decreased pain- like response to capsaicin, an effect that was reverted by replacement with 17β estradiol (E2) in ovariectomized rats [63].

Additionally, physiological scenarios such as the estrous cycle, have implications on TRPV1-mediated pain responses. Consistent with this, some reports have revealed that rats in the proestrus phase (high E2 concentrations) display higher capsaicin nocifensive responses than rats in the estrus phase (low E2 levels) [63]. Moreover, it has been reported that mechanical and thermal thresholds in female mice change during their estrous cycle, since lower thresholds to mechanical and thermal plantar stimulation are observed during the proestrus in comparison to mice in the estrus phase [64].

Remarkably, the TRPV1 knockout female mice exhibit similar thresholds to these noxious mechanical and thermal stimuli during the proestrus and estrus, suggesting that TRPV1 is a molecular effector of the differences in nocifensive behavior observed during the estrous cycle in wild-type mice [64]. 

In order to elucidate the molecular mechanism underlying genomic estrogen actions, experimental evidence has been put forward and it suggests that E2 up-regulates the levels of TRPV1 mRNA in a dose-dependent manner in the TG [65]. Additionally, Wu et al. have also reported that E2 significantly upregulates TRPV1 mRNA in the hippocampus from rat [66]. Similarly, mouse DRG neurons treated overnight with E2 revealed an upregulation of their TRPV1 mRNA levels, as compared to control cultures [64]. 

Moreover, an in vitro human sensory neurons model treated with E2 during 24 h, showed a significant increase in TRPV1 mRNA [67], an effect that was also obtained using a selective agonist of the ERβ [64,67]. Similarly, the positive effects of estrogen on TRPV1 expression were observed in non-neuronal tissues as synoviocytes, osteoclast precursors and endometrium [68,69,70]. Thus, these studies strongly suggest that estrogens positively influence TRPV1′s expression through a classical genomic pathway mediated by estrogen receptors, Figure 4, Table 1.

Consistent with the results described above, it was reported that the upstream regulatory genomic region (promoter) of the mouse Trpv1 gene, contains a putative estrogen response element (ERE) located between −3000 and 1000 pair bases of the transcription start site, suggesting that estrogens could directly regulate transcription of this gene through the interaction of estrogen receptors in the TRPV1 promoter sequence [71]. 

In contrast, lower levels of TRPV1 mRNA have been detected in different regions of the mouse brain during the proestrus, as compared to other phases of the estrous cycle [71]. Moreover, it was shown that estrogen differentially regulates the mRNA for various vanilloid receptors in the brain, suggesting a negative influence of this steroid on TRPV1 and TRPV5 mRNA but a positive effect on TRPV2, TRPV6 and TRPV4 mRNA levels during the proestrus, when estrogen levels are high [71].

The negative roles of estrogens on TRPV1 regulation have also been observed using long exposure to E2 or using an agonist of ERβ, since rat DRG neurons treated overnight with these compounds exhibited decreased capsaicin-induced TRPV1 currents without any effects on the total channel protein levels [72]. Intriguingly, a non-permeable E2 (steroid conjugated with bovine serum albumin (BSA)) was unable to affect TRPV1 function, indicating that E2 internalization is necessary in order to affect the function of TRPV1 [72]. This evidence is contrary to a previous report, where it was demonstrated that E2 potentiated capsaicin-induced current-densities in rat DRG neurons [73]. The difference between these results could rely on the concentration of E2 used in each assay since the negative role of E2 on capsaicin-induced currents was observed using a low concentration of E2 (100 nM) [72]. In contrast, potentiation of TRPV1 current-density was observed using supraphysiological concentrations of this steroid (100 µM) [73]. Thus, these contradictory studies suggest that E2 can positively and/or negatively modulate TRPV1 depending on the E2 concentration. Furthermore, these differences could be dependent on the ER expression in specific cells or specie, both of which are possibilities that remain to be explored.

Additionally to the nuclear effects dependent upon ERα and ERβ, E2 also produces its effects by interacting with the transmembrane receptor GPR30, which modulates estrogenic signaling from the plasma membrane. This pathway induces a fast response that occurs within seconds to minutes of receptor activation. Although the functional roles of the estrogen receptor involved in this mechanism are still unclear, it is likely that E2 acts through GPR30, triggering an intracellular signaling pathway that activates PKCε (Protein Kinase C epsilon) and inducing TRPV1 phosphorylation on serine 800 [74]. The phosphorylation of the channel in this residue decreases the threshold of TRPV1 activation [75] (sensitization) and it also plays an important role in regulating the association of the channel with tubulin, modifying microtubule-dependent pain sensitization, as reported by Goswami and collaborators [74,76]. Thus, E2 signaling through GPR30 positively regulates TRPV1 activity in a rapid fashion and shows that the estrogen plays a role as a sensitizer of the pain mediated by this channel, Table 1. 

Altogether, these findings suggest that estrogen actions converge on multiple signaling pathways, including genomic and non-genomic actions that affect the expression and function of TRPV1, Table 1.

### 2.2. TRPV1 Is also Targeted by Androgens

Initial reports showed that DHEA negatively modulates TRPV1 activation [73]. Assays in dissociated rat DRG neurons using electrophysiological whole-cell recording experiments, showed that DHEA decreases the capsaicin-evoked currents (EC_50_ = 6.7 µM) [73]. The activation of TRPV1 was negatively regulated by DHEA application to the neuron surfaces suggesting the existence of an extracellular putative steroid-binding site. Furthermore, using 3α-DHEA (a DHEA stereoisomer), it was shown that TRPV1 activation is upregulated, indicating that DHEA’s negative effects are stereospecific. Interestingly, testosterone, another androgen, is less effective at decreasing the activation of TRPV1, indicating that DHEA has specific structural requirements to exert its negative effects on TRPV1′s activation [73]. Even when the putative DHEA binding site on the TRPV1 channel remains undetermined, these results suggest a direct effect of this steroid on the modulation of this channel, Table 1. 

Contrary to the above, a positive role in TRPV1 activation by androstenedione has been recently reported. In this sense it was demonstrated that androstenedione activates currents in DRG neurons from the TRPA1 knockout mice and that these currents were inhibited by capsazepine [77], a TRPV1 antagonist, suggesting that these currents were due to TRPV1 activation, Table 1. It is interesting to note that some compounds used as treatments for breast cancer are based on their anti-estrogenic effects using compounds with similar structures to androstenedione and that they cause pain [78]. Although the molecular mechanism of TRPV1 activation by androstenedione is unknown, it is possible that this steroid and its synthetic analogs produce pain through direct TRPV1 activation, an aspect that requires more research efforts. 

Finally, it has also been reported that TRPV1 expression is negatively regulated by testosterone. It was demonstrated in a male rat model of orofacial myositis induced by CFA (Complete Freund’s adjuvant). The castration of these animals produced TRPV1 overexpression in the TG. Interestingly, TRPV1′s upregulation was not observed if the rats were treated with testosterone [79], Table 1. Thus, these data suggest that testosterone plays a negative role in TRPV1′s expression in this animal model and the molecular pathway for this type of modulation is an open field of research. 

### 2.3. A Non-Classical Progesterone Effect on TRPV1 Expression 

Recently, we reported that progesterone (P4) down-regulates TRPV1-protein levels through a non-classical pathway mediated by the Sigma-1 receptor (Sig-1R) [80]. This receptor has a role as a chaperone and is highly expressed in sensory neurons [81,82]. The function of Sig-1R is regulated by several compounds such as P4, which is an endogenous antagonist of Sig-1R’s function [83]. Sig-1R interacts with specific proteins and helps in their folding, traffic and stability. We identified that TRPV1 is a Sig-1R partner [80]. This TRPV1-Sig-1R interaction avoids the degradation of TRPV1 by the proteasomal pathway; thus Sig-1R has positive effects on the physiological role of TRPV1 as a transducer of nociceptive signals, Figure 5 and Table 1.

We demonstrated that Sig-1R/TRPV1 protein complex formation is interrupted by P4 or by synthetic antagonists of Sig-1R such as BD1063 [80]. As a consequence of this, TRPV1 protein is degraded and TRPV1-dependent pain thresholds are increased. In detail, we showed that pregnant mice, where P4 levels are high, capsaicin-produced pain-like behavior is decreased as compared to non-pregnant mice, suggesting that this physiological state in which there are high P4 levels, this steroid binds to Sig-1R and impedes the association of this chaperone with TRPV1 [80]. In agreement with this observation, it had also been reported by another research group that Sig-1R expressed in the central nervous system from pregnant female mice is unable to interact with synthetic agonists since high P4 levels lead to the occupation of the Sig-1R binding-sites [84]. 

The role of P4 as a disruptor of the interaction between Sig-1R and TRPV1 was also demonstrated by Cortés-Montero et al., [85]. Unlike our report, Cortés-Montero determined that Sig-1R interacts with the N- and C-terminus of TRPV1 in a fashion that was strongly dependent upon high concentrations of calcium, similarly as was shown for calmodulin and this same channel. The interaction of Sig-1R at the C-terminal region of TRPV1 is interrupted by P4, stimulating the calmodulin association [85]. 

These data showed that P4 blocks the association between Sig-1R and TRPV1, producing the downregulation of TRPV1 function by different mechanisms. First, the dissociation of TRPV1 and Sig-1R produces TRPV1′s destabilization and protein degradation [80]. Second, the dissociation of Sig-1R from the C-terminus of TRPV1 allows the interaction of CaM (calmodulin), which desensitizes the channel [85]. These molecular mechanisms have a common physiological consequence, a decrease of the pain produced by the activation of TRPV1, Figure 5 and Table 1. 

### 2.4. The Expression of the TRPV4 Channel Is Regulated by the Genomic Action of Progesterone 

TRPV4 channel was first identified as the vanilloid-receptor-related osmotically activated channel (VR-OAC), since the initial characterization of this protein showed that it is activated by hypotonicity [53]. TRPV4 is expressed in lung, spleen, kidney, testis, fat and also in sensory neurons and Merkel cells [53].

This channel is widely associated with channelopathies, including Charcot-Marie-Tooth disease type 2C, Spinal Muscular Atrophy and Hereditary Motor and Sensory Neuropathy type 2, among others [86]. 

The recent obtention of the TRPV4 high-resolution 3D-structure showed the typical assembly with a four-fold symmetry and the overall structural components common to TRP channels. Although, the pore domain of TRPV4 showed some particularities, suggesting that its gating is different from other members of the TRPV subfamily [87]. 

TRPV4 activators include warm temperature (~30–43 °C), and endogenously-produced arachidonic byproducts [49,50,88]. Strangely, few studies have focused on studying the regulation of TRPV4′s expression. However, it has been indeed described that TRPV4 is regulated by progesterone (P4) through a classical genomic pathway [89]. 

It was determined in epithelial cells from human airways and mammary glands with prolonged exposure to P4, that it caused downregulation of TRPV4′s mRNA and protein levels [89]. The P4 negative effects in TRPV4′s expression were reverted by an antagonist of the progesterone receptors, suggesting that P4 acts through a classical genomic pathway, Table 1. Furthermore, luciferase-reporter assays using cells transfected with a vector containing 1.3 kb of the promoter region of the human TRPV4 gene, displayed a decrease in luciferase activity when the cells were exposed to 100 nM of progesterone [89]. This result suggests that the TPRV4 promoter contains putative progesterone response elements, which are still undetermined [89]. 

Additionally, experiments carried out in human endometrial biopsies clearly showed that TRPV4 expression is downregulated during the luteal phase when P4 reaches high levels [90]. This report strengthens the negative role of this steroid on the regulation of TRPV4 expression and also demonstrates the propensity of TRP channels to be finely-regulated during the menstrual cycle. 

Furthermore, there is evidence of a fast-non-genomic influence of P4 on the activation of TRPV4 [91]. In mouse ciliated oviductal cells, which are important in the transport of eggs from the oviduct to uterus in several species [92], it was shown that P4 mediates this transport by modifying the velocity of the ciliary beat frequency. Recently, it was determined that P4 regulates calcium influx in this mouse ciliated oviductal cells, an effect that is facilitated through the activation of TRPV4 channels and is dependent on a protein complex formed between the GABA_A_ and GABA_B_ receptors (both proteins also co-expressed in the cells from the oviduct) [91], Table 1. 

The P4 interaction within GABA_A_ receptor produces the activation of GABA_B_ receptor, triggering the signaling through the G_i/0_ pathway and leading to IP3R activation (inositol triphosphate receptor) and the release of Ca^2+^ from the endoplasmic reticulum [91]. 

Finally, this pathway stimulates the TRPV4 sensitization [93], resulting in Ca^2+^ influx through TRPV4 and producing changes in the speed of the ciliary beat frequency [91]. 

Although it has not been experimentally demonstrated that P4 also has this effect in the respiratory airways, it is possible that this steroid and TRPV4, can also regulate the ciliary beat in this region, since the increase in intracellular Ca^2+^ is pivotal for the clearance of these airways [94].

### 2.5. Estrogens and TRPV5/6 Channels: Their Role in Calcium Homeostasis 

TRPV5 and TRPV6 channels, initially named as epithelial calcium channels 1 and 2, respectively [54,55,95], comprise channels of the vanilloid subfamily mainly expressed in the epithelia of the kidney and intestines. These channels conserve the overall structure of all the TRP channels, Figure 1, they can be assembled as homo or heteromeric channels and they are constitutively active [96,97]. 

The TRPV5 channel was isolated from rabbit kidney cell lines [95], whereas TRPV6 was identified in the rat duodenum as a protein sharing about 75% of homology with TRPV5 [55]. Their highest selectivity for Ca^2+^ characterizes these channels, pinpointing them as gatekeepers of apical calcium transport in specific tissues such as kidney and intestine [98]. The expression of these channels is also regulated by estrogens, as will be discussed. 

The first studies focused on characterizing the genes that encode for these channels in mice identified a putative estrogen response element localized at the upstream region of the Trpv6 gene [99]. Accordingly, the levels of TRPV5 and TRPV6 mRNA displayed upregulation in renal tissue and duodenum in ovariectomized rats with estrogen-replacement therapy [100,101]. The positive effects of estrogen on TRPV5/6 regulation were also observed in the aromatase knockout female mouse model, where the synthesis of estrogens is impaired and the female-mice display downregulation of TRPV5/6 gene expression, widely affecting renal calcium reabsorption [102], Table 1. 

Positive effects on TRPV6 expression mediated by estrogens have also been observed in the uterus from mice [103]. A treatment of 14-day-old female mice with E2 resulted in the upregulation of TRPV6 mRNA levels; this effect was reverted by the co-treatment with E2 and an antagonist of estrogen receptors, indicating that genomic actions of these receptors regulate TRPV6 expression [103]. It must be considered that the upregulation of a channel with a high affinity for Ca^2+^ could be crucial to uterine physiology, for example, affecting contraction and implantation of the embryo. 

Besides TRPV5/6 expression in peripheral tissues, these channels also are expressed in specific areas of the central nervous system [71,104]. Remarkably, it has been found that estrogens also influence the expression of these channels in the brain. For example, high TRPV6 expression is observed in the hypothalamus, olfactory bulb and cortex from female mice in the proestrus [71,104]. Furthermore, putative functional estrogen response elements were identified between -3000 and 1000 nucleotides from the transcription start site of the Trpv5 and Trpv6 genes, reinforcing the role of estrogens in regulating the TRPV5/6 expression through classic genomic effects [71].

Fascinatingly, TRPV5 has an essential role in the regulation of bone resorption by inhibiting osteoclasts, the cells that destroy the osseous tissue [70,105]. Thus, the reduction of estrogen levels in postmenopausal women is closely related to osteoporosis [106]. Interestingly, TRPV5 expression is downregulated in osteoclasts precursors from ovariectomized female mice, where estrogen levels are decreased [70], exemplifying that estrogens are required to positively regulate TRPV5 expression. Additionally, osteoclast precursors treated with E2 also show upregulation in TRPV5 and TRPV6 mRNA and protein levels, Table 1. Moreover, knockdown experiments using TRPV5 siRNA, have shown that TRPV5 depletion affects the inhibitory actions of estrogen on osteoclastogenesis [70]. At the molecular level, estrogens regulate TRPV5 expression in osteoclasts precursors through ERα actions that promote the binding of the subunits of the NFκβ (nuclear factor kappa-light-chain-enhancer of activated B cells) to the TRPV5 promoter region, leading to transcription [107], Figure 6.

The upregulation of TRPV5 expression stimulated by estrogens in osteoclasts delays osteoclast differentiation and stimulates apoptosis of these bone cells, decreasing bone resorption [107], Figure 6. Similarly, TRPV6 expression in osteoclast precursors is upregulated by estrogens [107] and TRPV6 knockout mice display osteopenia, indicating that this channel also has an important role in the control of osteoclastogenesis [108]. 

These studies have provided enough evidence of the importance of estrogens in modulating the expression of TRPV5/6 channels in the osseous tissue and their strong role in bone remodeling. These evidences have also highlighted these channels as targets in the treatment of postmenopausal osteoporosis. 

Finally, a non-genomic effect of estrogens in the regulation of the function of TRPV5/6 channels has also been reported. Using calcium imaging and whole-cell patch-clamp techniques it was revealed that E2 acts rapidly through a non-genomic pathway, increasing intracellular Ca^2+^ via TRPV5 and TRPV6 channels in kidney and colonic epithelial cells, respectively [109,110], Table 1; hence, this result suggests that TRPV5-6 activation by estrogens constitutes a fast mechanism for transepithelial Ca^2+^ reabsorption. 

TRPV5 expression is also regulated by other gonadal steroids, such as the androgens. This regulation plays an important role in sexual differences observed in urinary calcium excretion since it has been determined that male mice excrete more Ca^2+^ in urine than female mice [111]. The differential expression of proteins essential for calcium reabsorption between genders gives this variability. The TRPV5 channel is one of the proteins differentially expressed in the kidney with lower expression in male than in female mice [111]. Interestingly, gonadectomized male mice display decreased renal Ca^2+^ excretion while TRPV5 expression is increased; both effects are prevented with testosterone re-supplementation [111]. Although this result suggests that testosterone has a genomic effect that leads to a decrease in TRPV5 expression, preliminary experiments using a prostate cell line (which expresses the androgen’s receptor) transfected with a plasmid containing the TRPV5 promoter region cloned upstream of the luciferase reporter gene, did not display changes in the activity of luciferase in the presence of dihydrotestosterone [111], Table 1. Consequently, it is still necessary to deepen our knowledge of the molecular mechanism involved in the regulation of TRPV5 expression by androgens. 

## 3. TRPM Subfamily and Steroids

The melastatin-related transient receptor potential subfamily is constituted by eight channels (TRPM1-8) [112], Figure 1, and this subgroup is named after their first member was identified in melanoma cell lines (TRPM1) [113]. Although these channels possess their distinctive features, they also share the overall structure of all TRP channels. In particular, they have four TRPM high homology regions (MHR) located at their N-terminus [112], Figure 2, and a C-terminus that contains the TRP box and, some of these channels, also exhibit peculiar features such as the presence of NUDIX-like (TRPM2) [114] or kinase domains (TRPM6/7) [112] Figure 2. Most of these channels allow the flux of divalent cations; however, two members are impermeable to these ions (TRPM4/5) and instead they only allow the flux of monovalent cations [115], whereas TRPM6/7 have a preference for Mg^2+^ over Ca^+2^ ions and, they are the main regulators of Mg^2+^ homeostasis in the kidney [116]. An unusual feature of some TRPM channels is their voltage-dependence (TRPM3/5/8), even when they lack several of the positively-charged amino acids in their S4 [117]. Additionally, TRPM2/4/5, TRPM3 and TRPM8, are thermosensitive channels since they are activated by warm, hot and cold temperatures, respectively [118]. Similar to other TRP channels, TRPM members are activated in a polymodal fashion and some of them are directly regulated by steroids, as will be described below. 

### 3.1. TRPM3 Is the Main Target for the Effects of PregS

PregS has effects on the function of insulin-producing pancreatic cells and nociception. These roles are closely related to its regulation of the TRPM3 ion channel. 

TRPM3 was initially described as a Ca^2+^/Mn^2+^ permeable channel, which is expressed in the human kidney, brain, testis and spinal cord [119,120]. The identification of several isoforms of TRPM3 channels (six in humans and five in mice) has revealed that some of them are structurally different in their pore-forming region [121].

Among TRPM3-channel ligands, PregS is the canonical agonist for this channel [26]. Additionally, CIM021 is a synthetic agonist of the TRPM3 channel with higher affinity and potency than PregS [122]. Other TRPM3 activators are the D-erythro-sphingosine [123], and noxious heat [124]. 

PregS’s roles on TRPM3′s activation influences pancreatic β-cell physiology. Using TRPM3-transfected HEK293 cells and Ins1 cells (a cell line derived from rat pancreatic β cells), fluorometric Ca^2+^ imaging and electrophysiological experiments, it was determined that TRPM3 is activated in a reversible and concentration-dependent manner by PregS [26]. Moreover, actions of PregS on TRPM3 activation were also observed in insulin-positive mouse pancreatic islet cells where PregS potentiated glucose-induced insulin secretion [26]. Hence, although non-physiological concentrations of PregS (~35 µM) were necessary for this effect to occur, this report demonstrated the direct effects of PregS on the regulation of a TRP channel and the physiological role of TRPM3 in the function of pancreatic β cells [26], Table 1. Furthermore, activation of TRPM3 by PregS produces divalent cation influx. For example, Zn^2+^ ions are present in high concentrations in insulin exocytic granules and it has been shown that TRPM3, together with voltage-gated calcium channels, are primordial for Zn^2+^ uptake and for the maintenance of intracellular pools of this divalent ion in β cells [125]. 

Notably, TRPM3 activation is only produced when PregS is applied to the extracellular side of the plasma membrane [126,127]. This activation is strongly dependent on the cis-oriented sulfate group (or a negative charge at the same position), which suggests that there is stereo-selectivity for the interaction of PregS with the TRPM3 channel [126,127]. Moreover, this stereo-selectivity rejects the possibility that the effects of PregS are through altering membrane fluidity since a synthetic enantiomer of PregS is less capable of activating TRPM3 channels. These data suggest the existence of an extracellularly located binding site for PregS in the TRPM3 channel [126,127].

Besides the effects of PregS on pancreatic cells, the main role of this sulfate metabolite is as a neurosteroid, as mentioned above. PregS’s effects in the nervous system have been associated with a pro-nociceptive role and it occurs through different ways: by inhibiting the GABAergic and/or glycine transmission [128,129] and by potentiating the glutamatergic pathway [23]. Furthermore, PregS induces nociception through the activation of TRPM3 channels [26]. 

The use of PregS allowed for the description of the TRPM3 channel as a chemo- and thermosensor, which is highly expressed in small-diameter sensory neurons from DRG and TG [124]. Unlike a previous report, where it was described that the injection of 1.3 nmol (523ng) of PregS into the paws of rats had an antinociceptive effect in response to capsaicin since PregS reduces the capsaicin-evoked currents [130,131], the application of 2.5 nmol of PregS into the paws of mice elicited a nocifensive response [124], Table 1. Although PregS produced a less prominent paw-licking behavior than capsaicin, TRPM3 knockout mice lacked this pain response produced by PregS, whereas the capsaicin pain behavior was maintained [124]. 

The TRPM3 channel is activated by heat (≥40 °C) and PregS synergizes this activation [124]. This observation is relevant since low PregS concentrations (i.e., 500 nM) can produce TRPM3 activation in a pathophysiological scenario (i.e., inflammation) where the corporal temperature increases, leading to the generation of pain [124]. Moreover, TRPM3 activation by high concentrations of PregS (or CIM016) leads to neuropeptide release from isolated mouse hind-paw fresh-skin preparations, contributing to acute neurogenic inflammation, similar to what happens with TRPV1 and TRPA1 [122].

The roles of PregS as a regulator of TRPM3 channels in humans have been recently reported, demonstrating the expression of TRPM3 channels in human DRG where PregS also produces calcium influx through these ion channels [132], highlighting the relationship between this neurosteroid and TRPM3 channels in the nocifensive response. 

The first characterization of TRPM3 activation by steroidal compounds clearly showed that only PregS is an agonist of TRPM3 channels and that other steroids, such as pregnenolone, progesterone, DHEA, testosterone, are unable to activate this TRP channel [26]. However, some of these steroids can inhibit TRPM3 activation. For example, progesterone inhibits TRPM3 activity in response to PregS and nifedipine (a synthetic agonist of this channel), indicating that progesterone inhibits TRPM3 independently of the agonist [133], Table 1. Similarly, dihydrotestosterone is an inhibitor of TRPM3 activation; however, this inhibitory effect is only observed when TRPM3 is activated by PregS and not by another agonist, suggesting that dihydrotestosterone competes with PregS for a binding site in TRPM3 [133]. The existence of a steroid-binding site located in TRPM3 channels is supported by overlay assays that show that TRPM3 binds to PregS, progesterone and dihydrotestosterone, although TRPM3′s interaction with dihydrotestosterone is less effective in comparison to PregS [133]. 

Altogether, these results show that the TRPM3 function is regulated through a non-genomic role of PregS, modifying the excitability of the cell. Additionally, the effects of PregS on TRPM3 activation can be in turn modulated by other steroids, constituting a pharmacologic alternative to regulating the physiological effects of TRPM3 activation in pain responses. 

### 3.2. Aldosterone and TRPM6/7 Channels as Regulators of Mg^2+^ Homeostasis

TRPM6/7 channels have an exceptional structural feature, since their C-terminus exhibits the fusion of an α-kinase domain, Figure 2, and thus they are proteins with dual functions: ion channels and kinases (chanzymes) [134,135]. These channels preferentially permeate Mg^2+^ ions and are important regulators of the homeostasis of this cation. TRPM6 is highly expressed in the kidney and intestine, whereas TRPM7 is more ubiquitous [134,135]. Both channels can phosphorylate serine/threonine residues located in α-helices or coiled-coil structures from their target proteins (i.e., annexin-1). 

The association between TRPM6 and TRPM7 gives rise to functional heterotetramers, which are activated by changes in Mg^2+^ intracellular levels [136,137]. This protein complex is important for the appropriate reabsorption of Mg^2+^ from the kidney and it has been reported that the loss-function of TRPM6 channels is associated with severe hypomagnesemia [137]. Additionally, the reduction of Mg^2+^ reabsorption is also observed in aldosterone-infused mice [138]. These mice exhibit downregulation of TRPM7 expression and concomitantly, this produces negative effects on its downstream targets, such as annexin-1. Although, these mice display renal hypertrophy and inflammation, TRPM6 expression is unaffected [138]. Thus, high levels of aldosterone (hyperaldosteronism) alter the protein levels of TRPM7 expressed in the kidney, probably blocking the formation of the protein complex between TRPM6 and TRPM7, impacting on the reabsorption of Mg^2+^ at the distal tubule of nephrons and producing renal damage. 

The effects of aldosterone on TRPM6/7 channels are deeply dependent on the magnesium concentration, since aldosterone-treated mice that are Mg^2+^ deficient show decreases in the level of TRPM6 channels in the plasma membrane [139]. Consequently, these mice display severe kidney damage and hypertension, which is reversed by Mg^2+^ supplementation [139]. These data suggest that aldosterone negatively regulates the expression of TRPM6/7 channels, although the molecular mechanisms remain to be elucidated, Table 1. 

The actions of aldosterone on TRPM6/7 expression require several hours to be produced, suggesting that classical genomic effects of this steroid and its mineralocorticoid receptor could be at play. 

It is important to note that the α-kinase domain of TRPM7 is also essential to inhibit aldosterone signaling, which produces pro-inflammatory-related mediators such as ICAM-1, COX-2 and PAI-1 (Intercellular Adhesion Molecule-1, Ciclooxigenase-2 and Plasminogen activator inhibitor-1, respectively); thus, the α-kinase domain of TRPM7 is crucial to counteract the pro-inflammatory effects produced by aldosterone [139]. These data are in accordance with what is observed during renal injury and inflammation dysplayed in the mice treated with aldosterone where TRPM7 levels are downregulated [140]. Thus, the inflammatory actions of aldosterone are potentiated by the lacking of the α-kinase domain of TRPM7 which produces kidney dysfunction. 

In contrast to the report that showed downregulation of TRPM7 expression in the aldosterone-infused mice [138], experiments performed in TRPM7-expressing HEK293 cells demonstrated that the treatment of these cells with aldosterone raises the concentration of intracellular Mg^2+^ and that this phenomenon is highly dependent on the mineralocorticoid receptor and TRPM7 activation [140]. The increase in intracellular Mg^2+^ in TRPM7-expressing HEK293 cells is produced by augmenting TRPM7-protein levels at the plasma membrane and, in turn, increasing TRPM7 current-density in these cells [141], Table 1. The aldosterone-dependent changes in TRPM7-channel density in the plasma membrane required the actions of the mineralocorticoid receptors and the serum and glucocorticoid-regulated kinase 1 receptor (SGK1), since aldosterone is unable to increase TRPM7 levels in the membrane when inhibitors of these receptors are co-applied with aldosterone [141]. In addition, aldosterone is unable to produce increases in TRPM7 current-densities when this channel has a mutant-inactive α-kinase domain, indicating that the phosphotransferase role of TRPM7 together with the actions of aldosterone receptors are essential to transduce the signaling pathway associated with this steroid in order to produce changes in the levels of TRPM7 channels at the cell surface [141]. 

These data exemplify how a lipidic molecule, such as aldosterone, influences Mg^2+^ homeostasis modulating the expression of ion channels like TRPM6/7 through a classical pathway where the mineralocorticoid receptor is the main mediator for the effects of aldosterone. This evidence opens an area of research directed to elucidate the molecular mechanism by which the α-kinase domain of TRPM7 has an important role in aldosterone’s effects. 

Although aldosterone is the steroid best described for influencing TRPM6/7 channels, there is also experimental evidence showing that TRPM6 expression is regulated by E2 [142]. In this respect, ovariectomized female mice display downregulation of TRPM6 mRNA expression in the kidney. At the same time, the administration of E2 in these animals restores TRPM6 mRNA levels in this tissue [142], indicating that a gonadal steroid such as E2, positively regulates TRPM6 gene expression, an important component of renal Mg^2+^ homeostasis, Table 1. Accordingly, it has been demonstrated that postmenopausal women exhibit hypermagnesiuria which is significantly reverted by E2 replacement [143]. Thus, hormonal reestablishment with E2 has a magnesiotropic role, which probably causes changes in the expression of several protein targets, such as TRPM6.

Nonetheless, in general, it is still necessary to detail at the molecular level if the TRPM6/7 genes contain transcriptional regulatory elements important for the direct or indirect regulation by steroidal nuclear receptors or to search for other pathways involved in TRPM6/7 regulation by these types of steroids. 

### 3.3. TRPM8 as the Main Target of the Actions of Androgens 

The TRPM8 coding sequence was firstly isolated and cloned from a human prostate cDNA library [144] and, one year later, TRPM8 expression was also identified in sensory neurons from DRG and TG [145]. The structure of this channel was recently resolved and it was observed that it displays the overall structural features of TRP channels [146], Figure 1. Notably, these cation channels display some voltage-dependence, which is generated by the presence of an arginine located in the S4 [147]. The TRPM8 channel was identified as the menthol- and cold-transducer receptor, which is also activated by eugenol and the cooling agent icilin; furthermore, some lipid molecules also regulate its activation (i.e., steroids). 

TRPM8 channel has been described as a molecular marker of prostate cancer since it is overexpressed in prostate biopsies from patients with this type of malignancy. Moreover, the direct relationship between the TRPM8 channel and androgens was determined in prostate cancer biopsies from patients that received anti-androgenic therapy; in these samples TRPM8 gene expression was downregulated [148]. The establishment of TRPM8 as an androgenic gene was strengthened by experiments using a cell line responsive to androgens, the human prostate cell line LNCaP. 

The treatment of these cells with dihydrotestosterone produced upregulation of TRPM8 gene expression. An effect reverted by using an antagonist of the AR, which produces downregulation in TRPM8 expression [149,150]. Additionally, experiments with orchiectomized rats showed decreased levels of TRPM8 mRNA in the urogenital tract of these animals, an effect that was reversed by treatment with dihydrotestosterone, supporting the notion that TRPM8 is a primary androgen-responsive gene [151], Table 1.

Subsequent analysis of the regulatory genomic region of the TRPM8 gene showed that its promoter region has putative androgen response elements (ARE) [152]. By using chromatin-immunoprecipitation assays it was confirmed that AR binds to the AREs located close to the star transcription site of the TRPM8 gene [152]. All of these data have demonstrated that the classical genomic actions of androgens and their receptors (AR) regulate the TRPM8 gene expression, Figure 7A and Table 1. 

Prominently, testosterone shows high colocalization with TRPM8 channels in human prostate tissues and co-immunoprecipitation experiments using prostate cancer cells have revealed that testosterone is associated with TRPM8 localized in the plasma membrane [153]. The direct interaction of testosterone was assayed through overlay assays demonstrating that testosterone directly binds to purified TRPM8 protein and this interaction is specific since testosterone does not interact with the TRPV1 protein [153]. Furthermore, electrophysiological and image calcium experiments performed in primary human prostate cells revealed that testosterone induces Ca^2+^ influx through TRPM8 activation [153], Figure 7B and Table 1.

TRPM8′s activation by testosterone has also been observed in DRG and hippocampal neurons and in PC3 cells (a prostate cancer cell line which lacks AR expression) [154].

Unexpectedly, testosterone induces low Ca^2+^ influx in HEK293 cells that stably express TRPM8 [153]. The response of TRPM8 to testosterone in these cells was higher when the AR was downregulated using a specific siRNA or using an AR-blocker (hydroxyflutamide), indicating that ARs negatively influence the ionotropic effects of testosterone on TRPM8 channels [153]. The effects of testosterone on TRPM8 function were also assayed in planar lipid bilayers, where it was determined that testosterone (EC50 = 64.9 pM) and dihydrotestosterone (EC50 = 21.4 nM) produce TRPM8 activation, albeit with testosterone displaying a higher efficiency on TRPM8 activation, as compared to dihydrotestosterone [153]. Remarkably, the addition of purified AR into the bilayer system completely blocks the activation of TRPM8 by testosterone, indicating that both proteins could compete for the binding of testosterone. Finally, a testosterone binding site was proposed in the TRPM8 channel and suggested to be located extracellularly between S3 and S4, a region that contains several serine residues that are post-translationally modified and that are important for TRPM8 activation by several agonists. The change of these serines by glycine residues reduced the interaction of testosterone with TRPM8 and also decreased channel activation [153]. Although testosterone significantly reduces its interaction with the mutant channel at the plasma membrane, testosterone still binds to these channels in assays where total TRPM8 protein was used, indicating the existence of putative testosterone binding sites intracellularly in these channels which still need to be identified. 

The physiological role of TRPM8 regulation by androgens in prostate cancer cells has been linked to cell proliferation since the overexpression or activation of this channel in the PC3 cell line has an anti-proliferative effect [152,155], supporting the anti-tumorigenic role of TRPM8 channels [156]. 

Peculiarly, TRPM8 activation by testosterone in sensorial neurons has been related to the acute cooling sensation when this steroid is applied on human skin [153]. 

Furthermore and unexpectedly, it was also found that TRPM8 activation by agonists such as menthol or icilin was inhibited when the channel was previously exposed to testosterone [157]. TRPM8 inhibition was only observed when AR is co-expressed with the channel and when 10 nM testosterone, a concentration above the EC50 for TRPM8 activation (64 pM), was used, Figure 7C and Table 1. This effect involves another non-classical role of the AR, since the nuclear translocation is unnecessary to exert this effect. Moreover, it is required that the AR are available at the cytoplasm to interact with the cytoplasmic domain of TRPM8 channel [157]. This protein complex has an influence on cell migration, since PC3 cells co-expressing AR and TRPM8 channels exhibit a decrease in cell migration speed in comparison to control cells. Indeed, the exposure of these cells to 10 nM of testosterone, increased cell migration speed in comparison to cells co-expressing both proteins (TRPM8 and AR) and without testosterone treatment [157]. Interestingly, 10 nM of testosterone induces translocation of the TRPM8-AR protein complex to lipid rafts, where TRPM8 activity is inhibited and cell migration increased [157]. Indeed, a higher testosterone concentration, such as 100 nM did not affect the localization of the protein complex, although this concentration induced the dissociation of TRPM8 and AR without altering TRPM8 activation and PC3 cell migration [157]. 

Recently, a similar protein complex between TRPM8 and AR was reported in sensory neurons where it is an important mediator of differences in gender perception of cool temperatures [158]. This work shows that TRPM8 and AR are located at the cell surface of DRG neurons and that this protein complex is disrupted by the addition of 100 nM of testosterone, inducing AR translocation to the nucleus [158]. Furthermore, this concentration of testosterone inhibits the TRPM8 activation by menthol. Notably, castration of rats or mice increases their perception of cool temperatures, which was decreased by treatment with testosterone [158]. These results suggest that testosterone decreases sensitivity to cool temperature in sensory neurons through the inhibition of the protein complex between TRPM8 and AR at cell surface. 

The above experimental evidences support that androgens positively regulate TRPM8 channels through the classical genomic pathway and through a non-classical ionotropic effect of testosterone on TRPM8 function. Finally, a third unusual way to negatively regulate TRPM8 activity was observed when 10 nM of testosterone was used and it was found that TRPM8 was mobilized to caveolin-rich domains, Figure 7C, or by the use of 100 nM of this steroid in DRG neurons, which disrupted the protein complex between TRPM8 and AR. All of these pathways exemplify the influence of androgens on the expression, cell location and function of TRPM8.

## 4. TRPA1: Estrogens and Androgens Regulate Its Expression and Function 

The TRPA1 cDNA sequence was firstly isolated from human fibroblasts and it was shown to encode for a protein with a long N-terminus conformed by several ankyrin repeats (16 repeats in human), six transmembrane passes and a short C-terminus [159], Figure 1 and Figure 2. Later, this protein was identified in mouse where it was thought to be a TRP-like channel (ANKTM1) widely co-expressed with TRPV1 in small nociceptive neurons [160,161]. TRPA1 is activated by temperature (in a species-specific fashion [118]) and by organic compounds such as cinnamaldehyde and isothiocyanates found in cinnamon and mustard oil and wasabi, respectively [160,162]. Hence, this channel has an important function in the sensory system and it is associated with the transduction of noxious stimuli that produce pain responses [160,161,162].

The high-resolution 3D-structure of the human TRPA1 was recently resolved, displaying the typical homotetrameric assembly of a TRP channel. Particular TRPA1 structural features include a prominent cytoplasmatic domain, constituting 80 % of the total protein, a vast number of ankyrin repeats located at the N-terminus and an α-helix linker after the S6 that constituted an unpredicted TRP-like domain [163], Figure 2. 

TRPA1 activation by temperature has been controversial since some research groups have reported that TRPA1 is activated by cold [160,161], while others have reported that this channel is activated by noxious heat [164]. These opposing reports about TRPA1 activation by temperature have been associated with a species-specific context, since TRPA1 from invertebrates plays a role as a heat receptor and in mammals, this channel has been reported to act as a cold or heat thermosensor [164,165]. Interestingly, the TRPA1 channel has been described as a component of a “triad of channels that detect acute noxious heat” in mice [164], reasoning that murine TRPA1 is a heat-activated channel. Although the initial description of the human TRPA1 showed that it was activated by cold temperature [160], it also has been described that (in humans and primates) TRPA1 is a temperature-insensitive channel [165]. Finally, a recent report described that TRPA1 acts as a cold-sensor after exposure to a sensitizing stimulus, such as voltage or heat [166], suggesting that TRPA1 is a dual thermoreceptor. 

Besides the controversy on TRPA1 activation by temperature, several reports have revealed that this channel is regulated by reactive oxygen or nitrogen species that are endogenously-produced [167,168]. Likewise, this channel is activated by some lipid mediators which include lysophosphatidic acid, cholesterol [169,170], and steroids such as estrogens and androstenedione, as will be reviewed next. 

Estrogens are widely involved in endometriosis, causing debilitating chronic pelvic pain [171]. These lesions are highly innervated by afferent sensorial neurons co-expressing ion channels implicated in pain signal transduction, such as TRPA1 and TRPV1, among others, [171,172]. Furthermore, peritoneal biopsies from women with endometriosis display upregulation in the mRNA levels for TRPA1 and TRPV1 [67], indicating that the pain produced in this pathology could be associated with the overexpression of these pain mediators. The relationship between estrogens and TRP channels in the endometrium was also observed in female rats treated with a synthetic analog of estrogens, exhibiting augmented mRNA TRPA1 and TRPV1 levels [69], Table 1. Unexpectedly, the expression of these channels was detected in non-neuronal cells from rat endometrium, indicating that these channels also influence the physiology of endometrial cells. Since there is upregulation of these channels in both models (human endometrial biopsies and rat endometrial cells), this suggests that estrogens positively regulate the transcription of these genes and contribute to inflammation and pain states characteristic to endometriosis, Table 1. 

TRPA1 and TRPV1 gene expression in human endometria from healthy women confirmed that these channels are expressed in this non-neuronal tissue. Moreover, endometrium samples from women with the most severe form of endometriosis, such as deep infiltrating endometriosis, display higher TRPA1 and TRPV1 expression than the samples from healthy women [173]. The severity of the endometriosis (including symptoms as dysmenorrhea, dyspareunia and dyschezia) correlates with exacerbated expression of TRPA1 and TRPV1 channels, suggesting that these channels play a role in this disease. 

On the other hand, negative regulation of estrogens on TRP channel expression has also been reported since estrogen depletion induced by the ovariectomization of rats produced increases in TRPA1, TRPM2, and TRPV1 current-densities in primary hippocampal and DRG neurons; concomitantly, this produced accumulation of free cytosolic Ca^2+^ causing apoptosis of these cells [174]. This effect was reversed with E2 replacement therapy, suggesting that this steroid has a negative role in the function of these channels. Thus, estrogens have a protective role in these neurons since they counteract TRP channel activation (TRPA1, TRPM2 and TRPV1), avoiding cell death in response to Ca^2+^ overload [174]. Although the molecular mechanism involved in the negative regulation of estrogens on TRP channel activation remains to be further researched, this report suggested that some neurodegenerative diseases in postmenopausal women could be the result of the decrease in the levels of estrogens and excessive production of mitochondrial oxygen free radicals in these types of neurons [174]. 

The above-described studies exemplify the effects of estrogens on TRPA1 expression/function and the physiological results on sensory, hippocampal neurons and endometrial cells. 

Additionally, some metabolites of estrogens regulate the function of pancreatic β-cells modifying TRPA1 activity [175]. Firstly, it was demonstrated that catechol-estrogens (hydroxylated estrogens) increase glucose-induced insulin secretion in mouse pancreatic islets via an independent mechanism of the estrogen receptors. This effect was also observed in INS-1 cells (rat insulinoma cell line), which express the TRPA1 channel. Interestingly, this effect was reverted by the co-application of catechol-estrogens and selective blockers of TRPA1 channels [175]. 

Furthermore, experiments, where TRPA1 was knocked down in INS-1 cells, showed that catechol-estrogens are unable to produce their effects in these cells, which strongly indicated that these hydroxylated estrogens increase glucose-induced insulin secretion in a TRPA1-dependent fashion [175], Table 1. Finally, experiments carried out in TRPA1-expressing HEK293 cells revealed that catechol-estrogens directly induce TRPA1 activation. Whole-cell recordings demonstrated that 2-hydroxy-estrone (a catechol-estrogen) evoked TRPA1 currents similar to a well-characterized TRPA1 agonist, cinnamaldehyde; thus, this metabolite of estrogens is considered a novel endogenous agonist for this TRP channel. This evidence is another example of the non-genomic and pleiotropic effects of steroids and their byproducts, although the detailed molecular mechanism remains to be determined [175].

Further studies on the non-genomic effects of steroids affecting TRPA1 function have been performed and it has been reported that androgens affect its activation. The first evidence was obtained using synthetic aromatase inhibitors [78], these compounds hinder the conversion of androgens to estrogens and are widely used in the treatment of estrogen-positive breast cancer in postmenopausal women. However, these molecules produce severe pain as a collateral effect [176,177]. Due to the electrophilic nature of these compounds and their association with pain production, it was thought that they produced pain through a TRP channel: TRPA1. Calcium imaging assays and whole-cell patch-clamp recordings carried out in HEK293 cells expressing human TRPA1 channels and in mice DRG neurons demonstrated that these compounds produced TRPA1 activation with the consequent Ca^2+^ influx and current generation through TRPA1 channels [78]. Furthermore, these compounds did not produce Ca^2+^ influx in DRG neurons from knockout TRPA1 mice. In addition, it was found that aromatase inhibitors produced pain-like behaviors in mice which were not observed in the mice lacking TRPA1 expression. These results suggest that aromatase inhibitors are agonists of TRPA1, although a possible molecular mechanism for these effects has not been defined [78]. 

Aromatase inhibitors share some structural features with androstenedione; moreover, the plasma levels of this androgen increase after the treatment with these inhibitors. Thus, a possible synergic effect between androstenedione and the aromatase inhibitors to produce pain was investigated. Whole-cell electrophysiological experiments carried out in HEK293 cells expressing the human TRPA1 channel showed that the application of androstenedione elicited inward currents, which were abolished by a selective TRPA1 blocker [77], Table 1. It was also assayed in mouse DRG neurons, where androstenedione produced calcium influx partially dependent on TRPA1 since DRG neurons from TRPA1 knockout mice displayed residual currents which were abrogated by capsazepine, as was mentioned above [77]. Although androstenedione is unable to directly activate TRPV1 channels expressed in HEK293 cells, in the native system of expression of these TRP channels (DRG neurons), it is likely to contribute to Ca^2+^ influx. This possibility was confirmed in HEK293 cells co-expressing both channels where Ca^2+^ influx in the presence of androstenedione was partially decreased by a TRPA1 blocker or capsazepine and abolished when both inhibitors were co-applied [77], Table 1. Thus, these channels cooperate to exert the effects of androstenedione when they are co-expressed, as would be expected in a subgroup of small sensory neurons. Although androstenedione did not induce any acute-pain behavior in mice injected in their paws with this steroid, 30 and 120 min after its application, the mice displayed mechanical allodynia that was completely reversed by a TRPA1 blocker and partially by capsazepine [77]. Additionally, it was observed that low concentrations of androstenedione, aromatase inhibitors and H_2_O_2_ cooperatively produced mechanical allodynia, an effect that was entirely dependent on the TRPA1 channel, since it was not present in mice lacking TRPA1 expression [77]. 

These results suggest that androstenedione positively regulates TRPA1 activity; however, if this effect is through a direct interaction of the steroid with a binding site in TRPA1 or if it occurs through a signaling pathway still remains to be elucidated, Table 1. Nonetheless, there are some clues: it has been reported that HEK293 cells expressing a mutant TRPA1 channel (TRPA1 3C mutant,) which displays decreased responses to electrophilic agonists [178], this mutant channel is insensitive to androstenedione effects suggesting that this steroid requires specific amino acids located at the N-terminal region of TRPA1 that are mutated in the TRPA1 3C version (C619S, C639S, C663S and K708Q) of the channel. This result suggests a possible direct effect of androstenedione on TRPA1 activation [77]. 

## 5. A Brief Structural View of the Steroid Pocket in Some TRP Channels

Although some studies suggest that certain steroids have a role as ligands of specific TRP channels, it still necessary to perform studies that detail the putative steroid-binding sites located in the TRP channels. Most of the studies that demonstrated the ionotropic effects of the steroids were performed when the structures of some TRP channels were not resolved; thus, the putative binding sites were difficult to predict. However, now, most of the structures of TRP channels have be obtained by single-particle cryo-EM (with exception of the TRPM3 channel), facilitating the prediction of steroid binding sites.

For example, we have reviewed that androstenedione evokes currents through TRPA1 and TRPV1 channels [77]. Although, there is information suggesting an androstenedione binding site located in the N-terminus of the TRPA1 channel [77], complementary information can be obtained performing molecular docking of androstenedione and TRPA1 or TRPV1 channels, as we have done below. 

Figure 8A shows a docking simulation displaying that androstenedione interacts in a pocket formed between the N- and C-terminus of the TRPA1 channel. This pocket contains the residues E558, H589, N590, K591 and R592 located in the linker domain before the pre-S1 (in the N-terminus) of TRPA1 [163]. This pocket also comprises the residues Y1049, R1050, K1052, N1053 and F1056 situated in the coiled-coil structure at C-terminus of TRPA1 channel (Figure 8A), where the binding of other lipids such as the inositol hexakisphosphate (IP6) has been reported [163].

The androstenedione pocket in the TRPA1 channel does not share residues with the binding site identified in the aromatase (the enzyme that binds androstenedione to covert it in estrone) [182]. Androstenedione interacts with this enzyme through hydrogen bonds (H-bond) with residues M374 and T310; also, this steroid produces hydrophobic interactions with several residues such as R115, F134, F221, W224, I305, A306, D309, V370, V373, M374 located in the aromatase [182].

Our docking does not exhibit the formation of any H-bonds nor of hydrophobic interactions between androstenedione and TRPA1; thus, it could be relevant to determine if the residues displayed by the docking of androstenedione and TRPA1 are functionally relevant.

Likewise, we performed molecular docking to predict the binding of androstenedione to the TRPV1 channel, Figure 8B. Interestingly, the prediction shows that androstenedione binds in the C-terminus of TRPV1, contiguously to the TRP box; furthermore, androstenedione forms a H-bond with the R717 residue; in addition, this pocket comprises residues K714, A719, M716 (some of them could establish hydrophobic interactions with the steroid, as it happens in the aromatase), Figure 8B. 

The docking also shows some residues such as I387, T389, C390. L397 and A400 located at the linker domain before the pre-S1 of TRPV1 as part of the androstenedione binding-pocket, Figure 8B.

Interestingly, androstenedione binds near to a domain implicated in the gating of TRP channels, which it is also the binding-site for some positive regulators of TRPV1 and TRPA1 channels. These possibilities highlight the relevance of detailing the functional regulation of TRP channels by steroids.

This review has also discussed the ionotropic effects of testosterone through its interaction with the TRPM8 ion channel. Although there is evidence suggesting that testosterone binds extracellularly and intracellularly to TRPM8 channel [153], it is still necessary to detail the structural requirements of this interaction. 

Until now, the structures resolved for TRPM8 are from the avian sequence, which has high homology to the human sequence [146]. Thus, the possible steroid binding sites predicted in the avian structures can be extrapolated to the human TRPM8 channel. The molecular docking simulation of testosterone and TRPM8 reveals that the steroid is inserted in a pocket formed by residues located in the pre-S1 (L669 and W673), S1 (I687), and S2 (F726, F729, V733), Figure 8C. These residues form a hydrophobic interaction with testosterone. Furthermore, the docking displays a H-bond between testosterone and R988 located at the TRP-box of this channel, Figure 8C.

This pocket shows similitude to the cavity used by testosterone in the aromatase (an enzyme that also binds testosterone to convert it into 17β estradiol) [183]. Additionally, it has been reported that testosterone binds to the AR through a H-bond with an arginine (R752) [184], similarly to the H-bond predicted for the interaction with the TRPM8 channel. These reports strengthen the idea that the pocket for testosterone predicted in the TRPM8 channel could be functional and that it is located between the preS1 and S1-S2, where testosterone could establish hydrophobic interaction with these regions. Additionally, testosterone contacts the TRP-box of the channel through a H-bond with the R988 residue. These predictions open the possibility to perform site-directed mutagenesis and to evaluate the effects of substituting these residues in order to determine their functional relevance in the direct activation of TRPM8 by testosterone. 

## 6. The Relationship between the Thermal Threshold of TRP Channels and Steroids

This review also has described several evidences suggesting that some steroids, such as PregS and DHEA, regulate the activation of thermo-TRP channels, negatively modulating TRPV1 currents or such as androstenedione, a potential agonist of TRPA1 and TRPV1 channels [77,130,131]. However, the only experimental evidences demonstrating the direct action of some steroids on TRP channel activation are testosterone and PregS, which modify the function of TRPM8 and TRPM3 channels, respectively [26,185]. These examples represent the clearest evidence of how some steroids acts as ligands of TRP channels since they directly interact with them. 

These observations lead to the following question: Does the sensing of temperature by TRP channels change when they form complexes with steroids? This is a question that definitely requires further work. The interaction of TRPM3 with PregS is an excellent example of the interplay between steroid, TRP channel and changes in the temperature threshold [124]. PregS and heat show synergism on TRPM3 activation, resulting in a phenomenon where low concentrations of the steroid (0.5 µM) and 37 °C strongly activate TRPM3, while the concentration of steroid alone required to activate TRPM3 currents is ~40 µM, indicating that PregS is an allosteric modulator of TRPM3′s activation by temperature. 

Despite the lack of experimental evidence related to thermal activation of other TRP channels when steroids are present, some interesting hypotheses can be made. For example, it has been reported that cholesterol depletion in HEK293 TRPM8-expressing cells disrupts lipid rafts where TRPM8 is located, leading to a more efficient activation of TRPM8 by menthol or temperature, as compared to control cells [186]. Remarkably, the threshold for activation by cold temperatures in cholesterol depleted cells shifts to warm temperatures, indicating that dislocation of TRPM8 from lipid rafts modulates its temperature sensory proprieties [186]. 

Contrary to what happens with cholesterol depletion, 10 nM of testosterone maintains the TRPM8/AR protein complex in the lipid rafts [157], inhibiting TRPM8 activation by menthol and probably maintaining the temperature threshold to activation without changes (i.e., in the cool range). However, high testosterone levels (100 nM) disrupt the association between TRPM8 and AR, releasing TRPM8 from lipid rafts and favoring its activation by menthol [157]. Hence, we could propose that 100 nM of testosterone changes the temperature threshold for TRPM8 activation, shifting it to a warmer range, similarly as happens with cholesterol. 

Recently, it was reported that carboxamide steroids (synthetic compounds) inhibit the opening of TRPV1 and TRPA1 and that this negative regulation is through disruption of the lipid rafts [187]. Although this last study does not assess changes in the temperature threshold of activation of these channels, it will be interesting to evaluate this possibility. Besides, it could be very interesting to determine possible changes in the physical or biophysical properties of these thermo TRP channels complexed with steroids and if there are changes in the dependence of their activation by temperature. 

## 7. Conclusions

Since the discovery of the first mammal TRP protein to the obtention of the high-resolution structure of some of these channels, there has been an extensive effort to understand the pivotal role that these proteins play in several physiological events. 

The polymodal activation of these channels allows for the regulation of their function through different ways, including the actions of several synthetically and endogenously produced compounds that can directly bind to the channels to modulate their activity or/and regulate their expression. Interestingly, endogenous steroids have been shown to regulate these transmembrane proteins using both mechanisms [11], a phenomenon that represents how hydrophobic molecules regulate their function. 

Several of the well-known effects of steroids happen through the regulation of TRP channel expression, importantly contributing to pain, bone homeostasis, kidney function, insulin secretion, prostate cancer, among others, Table 1. 

For example, PregS and androstenedione have roles as “algogenic” molecules, since they positively regulate the expression/function of some of the master transducers of noxious stimuli: TRPM3, TRPA1 and TRPV1 channels, a “triad of receptors” recently identified as the main transducers of pain associated to noxious heat [164]. Additionally, this regulation constituted a possible molecular mechanism to explain sexual dimorphism of pain, since these steroids could differentially modulate the threshold to pain depending on their abundance in each gender. 

This sexual dimorphism is also observed in renal reabsorption, since males secrete more Ca^2+^ than females, and this effect is mediated by the negative regulation of TRPV5 expression by androgen’s actions. Possibly, postmenopausal osteoporosis in women is produced by augmented levels of androgens, leading to downregulation of TRPV5 and increasing the loss of renal Ca^2+^ and bone mass. 

Estrogens are also important mediators of bone homeostasis and part of this effect is also through the regulation of TRPV5 channels; hence, the interplay between these channels and estrogens maintains the balance between resorption and bone formation. This complicates the scenario in postmenopausal women since the reduction of estrogens with the concomitant increase in androgens leads to downregulation of TRPV5 channel expression contributing to osteoporosis by affecting bone homeostasis and Ca^2+^ reabsorption. Thus, estrogen replacement therapy is an alternative that restores the expression and function of TRPV5 channels and alleviates postmenopausal osteoporosis. 

Moreover, the relationship between estrogens and the TRPV5-6 and TRPM6 channels positively influences renal function favoring reabsorption of Ca^2+^ and Mg^2+^, respectively. Thus, a decay in estrogen levels also affects renal function. Similarly, aldosterone´s actions regulate the expression of the TRPM6-7 channels in the kidney. These observations exemplify how some steroids and TRP channels play pivotal roles in renal electrolyte balance and variations in the relationship between them strongly affect renal physiology. 

Most of the steroidal effects on TRP channels mentioned above are through classical genomic effects. However, some steroids regulate TRP channels through their non-classical effects that include proteinic intermediaries. An example of this is Sig-1R, a chaperone that is a target of P4. When P4 binds to Sig-1R, it inhibits its chaperone function, hindering the stabilization of the TRPV1 protein by Sig-1R. This condition, in turn, downregulates the TRPV1 protein increasing the pain threshold associated with this channel. Thus, progesterone indirectly acts as a protective steroid in the face of certain painful stimuli. 

There are indeed exceptional cases where these hydrophobic molecules act as ligands of TRP channels directly modifying their function. For example, the ionotropic effects of PregS and testosterone on TRPM3 and TRPM8 channels, respectively, represent novel molecular pathways for steroidal actions, functioning as endogenous ligands of these channels. These actions are faster than the genomic effects, thus these steroids modify the physiology of the cells where TRP channels are expressed. An example of this is observed in pancreatic β cells, where glucose-stimulated insulin secretion increases when TRPM3 and TRPA1 are activated by PregS and catechol-estrogens, respectively. Likewise, testosterone activates TRPM8 through a direct interaction of the steroid with the channel, modifying migration in human prostate cancer cell lines and the cold perception in mammals. 

In conclusion of all the studies described here highlight the importance of the regulation of the TRP channel function by steroids for the physiology of the neural, renal and osseous tissues. 

## Figures and Tables

**Figure 1 ijms-21-03819-f001:**
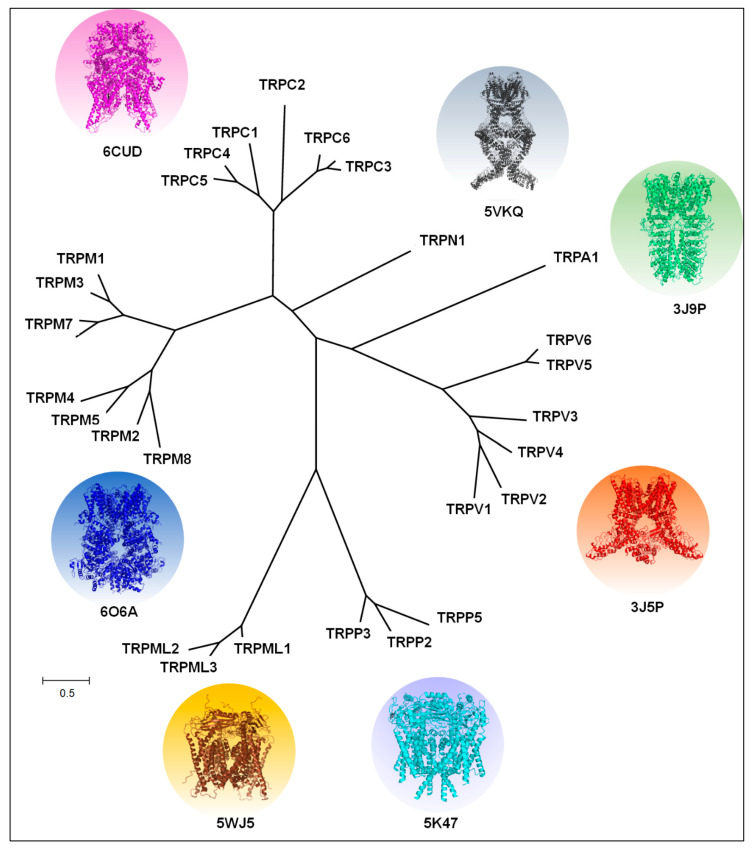
Phylogenetic tree of human transient receptor potential (TRP) channels. Protein sequence homology analysis shows the six TRP subfamilies present in mammals and the TRPN subfamily (presented in lower vertebrates). Since TRPC2 is a pseudogene and TRPN1 is not expressed in humans, the sequences used for these two channels were from mice and fish, respectively. The phylogenetic analysis shows the relations between the subfamilies. Protein sequences were aligned using ClustalW2 at the EMBL-EBI server. The scale bar represents 0.5 substitutions. Circles show ribbon representation of the 3D-structure for some TRP channels: TRPV1 (3J5P), TRPA1 (3J9P), TRPM8 (6O6A), TRPC3 (6CUD), TRPML1 (5WJ5), TRPN1 (5VKQ) and TRPP2 (5K47).

**Figure 2 ijms-21-03819-f002:**
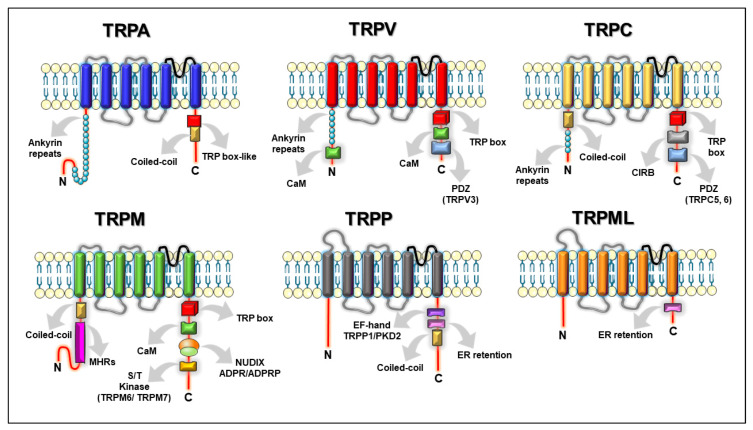
Structural domains and motifs in the N- and C- terminus of TRP channel subfamilies. Structurally TRP channels are formed by six transmembrane domains (S1–S6) with the pore loop between S5 and S6; four identical subunits form a functional channel. The N- and C-terminus (N and C) are located intracellularly and show some well-recognized domains and motifs that are involved in assembly, activation and regulation of these channels. The TRP box is a conserved domain common to the members of TRPC, TRPV, TRPM subfamilies and a TRP-Box like region is also found in the TRPA1 channel. Coiled-coil domains (CC) are positioned in the C-terminus of some TRP channels. TRPM channels share melastatine high homology regions in their N-terminus, (MHRs). Binding domains for calmodulin (CaM) and inositol triphosphate receptor (InsP3R) binding site (CIRB) as well as the PDZ-binding specific motif (PDZ) are present in the TRPC subfamily. The adenosine 5’-diphosphoribose (ADPR) or the ADPR-2’-phosphate (ADPRP), substrates of NUDT9 domain (a sequence motif that share a homologous region with the C-terminus NUDIX box), underly the gating properties of TRPM2 channels in the presence of ADPR. TRPM6 and TRPM7 channels present a serine/threonine kinase domain (S/T kinase). TRPP1 or PKD2 presents a calcium-binding motif (EF-hand), while TRPP and TRPML characteristically exhibit an endoplasmic reticulum retention (ER retention) domain in their C-terminal regions.

**Figure 3 ijms-21-03819-f003:**
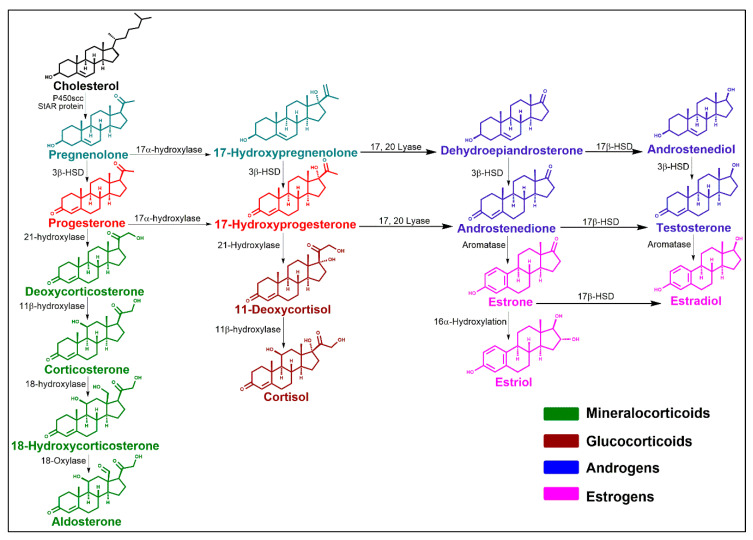
Representation of steroidogenesis. Schematic representation of the steroidogenesis pathways. Steroidogenic enzymes are depicted in black arrows. The main steroidogenic pathways are colored in green for the mineralocorticoids, brown for the glucocorticoids, blue forandrogens and pink for the estrogens’ synthesis.

**Figure 4 ijms-21-03819-f004:**
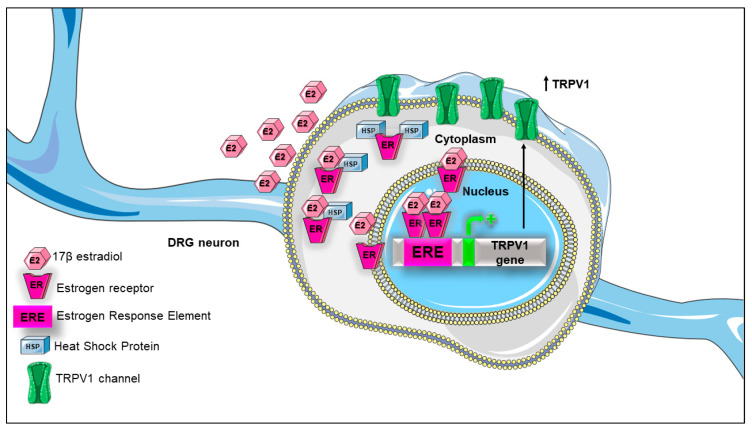
Classical genomic effect of estrogen in TRPV1 gene expression. Estrogens such as the 17β estradiol (E2) diffuse through the lipid bilayer of dorsal root ganglion (DRG) neurons and binds to the estrogen receptors (ER) located at the cytoplasm. This induces the dissociation of the ER and the heat shock proteins (HSP), allowing the translocation of the ER to the nucleus, where it binds to the estrogen response elements (ERE) located in the promoter region of the TRPV1 gene. This turns on the transcription of the TRPV1 gene, exacerbating its expression in DRG neurons and enhancing the pain state.

**Figure 5 ijms-21-03819-f005:**
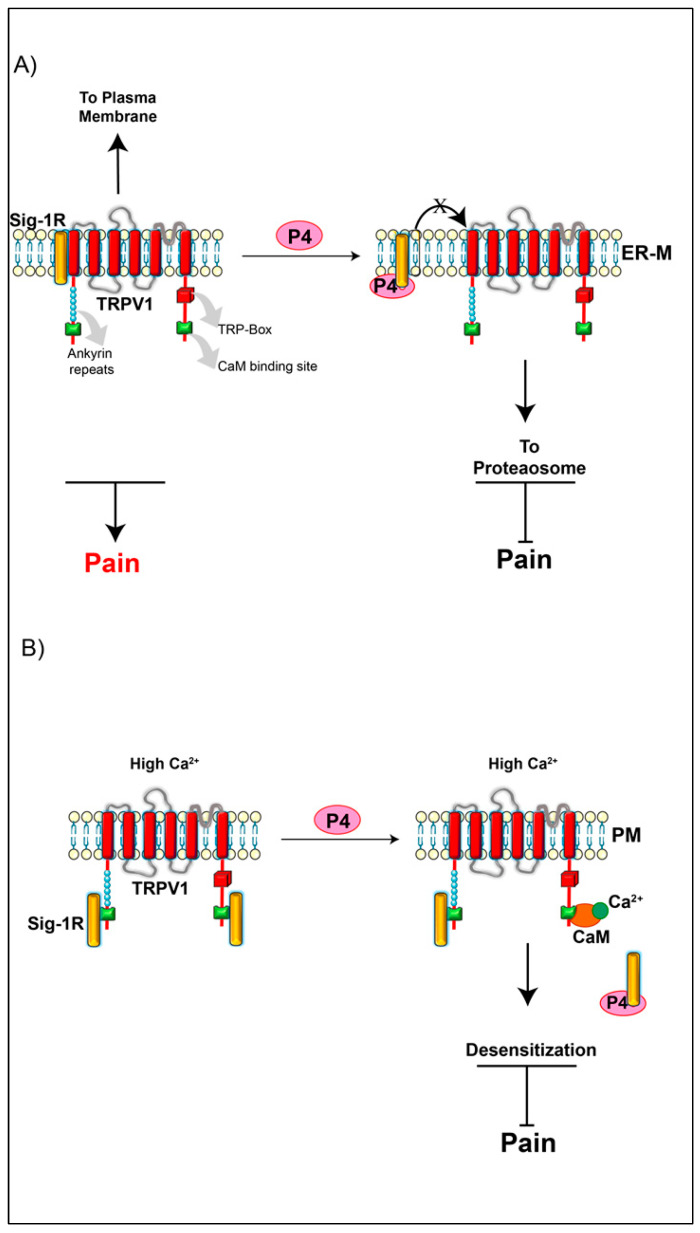
TRPV1 is negatively regulated by progesterone producing pain relief (**A**) The Sig-1R/TRPV1 protein complex positively regulates TRPV1 protein levels and the pain produced through this channel (left). Progesterone (P4) binds to Sigma-1 receptor (Sig-1R), destabilizing the protein complex between these proteins, producing TRPV1 degradation through the proteasomal pathway and downregulation of pain (right). (**B**) Sig-1R interacts with TRPV1 in the calmodulin (CaM) binding sites (left). P4 binds to Sig-1R, producing the dissociation of Sig-1R from the CaM binding site located at the C-terminus of TRPV1, in turn, it allows the interaction of CaM in this site producing channel desensitization (right).

**Figure 6 ijms-21-03819-f006:**
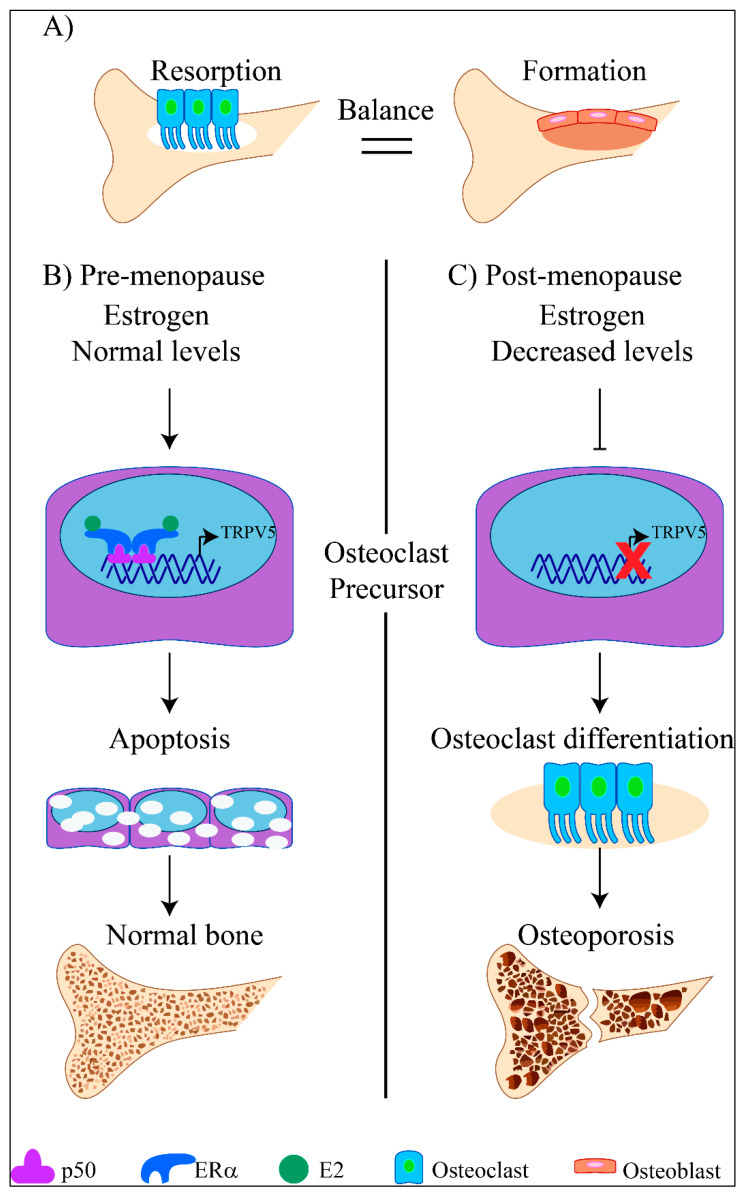
Regulation of osteoclastogenesis by estrogens and TRPV5 channels. (**A**) Bone homeostasis is maintained by the equilibrium between bone resorption and bone formation. (**B**) Estrogens levels in premenopausal women positively regulate TRPV5 expression, since estrogens receptors (ERα) facilitate the binding of p50 (a subunit of the NFκB) to the TRPV5 promoter region, leading to TRPV5 transcription, inhibiting osteoclast differentiation and inducing the apoptosis of these bone cells. These effects allow the balance between bone resorption and formation. (**C**) The reduction of the estrogen levels in postmenopausal women decreases TRPV5 expression, inducing osteoclast differentiation; thus, bone resorption is upregulated, producing postmenopausal osteoporosis.

**Figure 7 ijms-21-03819-f007:**
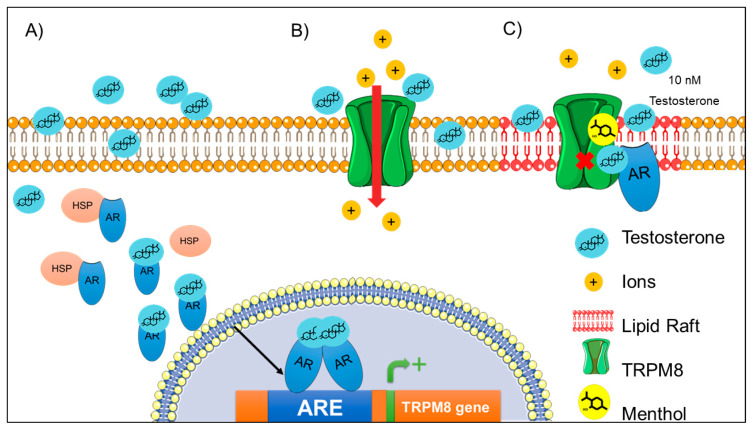
TRPM8 channels are regulated by androgens through different mechanisms. (**A**) Classical genomic effect of testosterone. Testosterone binds to the Androgen Receptor (AR), producing the dissociation of the protein complex between AR and heat shock proteins (HSP). This allows the translocation of the AR to the nucleus. AR recognizes the Androgen Response Element (ARE) in the promoter region of the TRPM8 gene, resulting in the upregulation of TRPM8 gene transcription. (**B**) Ionotropic effect of testosterone. Testosterone binds extracellularly to TRPM8 ion channels located at the cell surface producing their activation and leading to cation currents through the channels. (**C**) Non-classical action of AR by direct interaction with TRPM8 in the plasmatic membrane. AR and TRPM8 channels form a protein complex. Testosterone (10 nM) induces the translocation of the TRPM8-AR complex to lipid rafts, inhibiting TRPM8 activation by agonists such as menthol.

**Figure 8 ijms-21-03819-f008:**
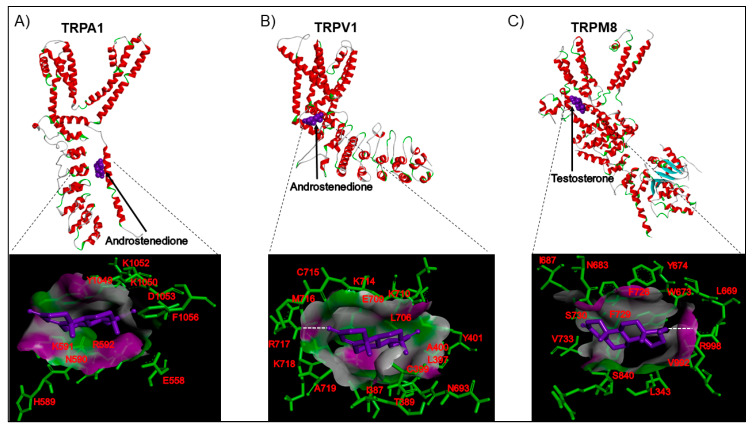
Predicted steroids bindings pockets localized in some TRP channels. (**A**) Molecular docking analysis of androstenedione with the TRPA1 channel (PDB 3J9P), (**B**) TRPV1 channel (3J5P) and (**C**) testosterone with the TRPM8 channel (6O6A). H-bonds are indicated by the dotted white line. The molecular dockings were performed using the Autodock-Vina software [179,180,181] and the image was prepared in Discovery Studio Visualizer.

**Table 1 ijms-21-03819-t001:** Genomic and non-genomic steroidal actions on TRP channel expression and function.

Steroid	TRP Channel	Type of Steroid Action	Overall Effect	Physiological Effect
**Estrogens**				
	TRPV1	(+) genomic (ERα/β)	Upregulation in TRPV1 expression	Pain induction Endometriosis Bone resorption
		(+) non-genomic	Potentiation of capsaicin-evoked currents	Undetermined
		(+) non-genomic (GPR30)	TRPV1 phosphorylation	Pain-Sensitization
		(−) non-genomic (ERβ)	Decrease in capsaicin-evoked currents	Undetermined
		(−) genomic	Low TRPV1 mRNA levels	Undetermined
	TRPV5/6	(+) genomic (ERα/β)	Upregulation of TRPV5/6 expression	Ca^2+^ renal reabsorption Bone homeostasis
		(+) non-genomic	Ca^2+^ influx	Ca^2+^ renal/intestinal reabsorption
	TRPM6	(+) genomic	Upregulation in TRPM6 expression	Mg^2+^ renal reabsorption
	TRPA1	(+) genomic (ERα/β)	Upregulation of TRPA1 expression	Endometriosis
Catechol-estrogens	TRPA1	(+) non-genomic	TRPA1 activation	Glucose-induced insulin secretion
**Androgens**				
Testosterone	TRPV1	(−) genomic	Downregulation of TRPV1expression	Decrease in pain
DHEA	TRPV1	(−) non-genomic	Decrease in capsaicin-evoked currents	Undetermined
Testosterone/DHT	TRPV5	(−) non-genomic	Downregulation of TRPV5 expression	Renal Ca^2+^ excretion
	TRPM8	(+) genomic (AR)	Upregulation in TRPM8 expression	Anti-tumorigenic effect in prostate cancer cells
Testosterone/DHT	TRPM8	(+) non-genomic (direct binding to TRPM8)	TRPM8 activation	Anti-tumorigenic effect in prostate cancer cells Cooling sensation
Testosterone	TRPM8	(−) non-genomic (AR and 10 nM testosterone)	TRPM8 inactivation	Increase in cell migration speed
DHT	TRPM3	(−) non-genomic	Decrease of TRPM3 activation by PregS	Undetermined
Androstenedione	TRPA1	(+) non-genomic (possibly direct binding to TRPA1)	TRPA1 activation	Mechanical allodynia
	TRPV1	(+) non-genomic (effect dependent of TRPA1)	TRPV1 partial activation	Mechanical allodynia
**Progesterone**				
	TRPV1	(−) non-genomic (Sig-1R)	Downregulation of TRPV1 protein levels	Increase in pain threshold
	TRPV4	(−) genomic (PR)	Downregulation of TRPV4 expression	Ciliary beat frequency
	TRPV4	(+) non-genomic (GABA receptors)	Increase in Ca^2+^ influx	Ciliary beat frequency
	TRPM3	(−) non-genomic	Decrease of TRPM3 activation by PregS and nifedipine.	Undetermined
**Pregnenolone sulfate (PregS)**				
	TRPM3	(+) non-genomic (direct binding to TRPM3)	TRPM3 activation	Glucose-induced insulin secretion Nociception
	TRPV1	(−) non-genomic	Decreased capsaicin-evoked currents	Antinociceptive effect
**Aldosterone**				
	TRPM6/7	(−) genomic	Downregulation of TRPM6/7 expression	Renal physiology
	TRPM7	(+) non-genomic	Increase in TRPM7 channels located at plasma membrane	Undetermined

(+)/(−) refers to positive or negative action of the steroid on TRP channel regulation. The word in parenthesis indicates the effector of the action of a particular steroid.

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
