# Peer review of "Steroids and TRP Channels: A Close Relationship"

_ijms, 2020, doi:10.3390/ijms21113819_

Round 1
Reviewer 1 Report
The manuscript aims at reviewing the TRP-channels and their interactions with different steroid molecules, in a rather comprehensive manner. The structure of this review is appropriate, and it is supplemented with an adequate number of tables and figures.
- My biggest concern is the novelty of this paper, in particular, given the broad interest in TRP channels, I dearly miss information on how the current review differs from hundreds of other reviews on this topic being published all the time. The authors should clearly define their manuscript, stating what new perspectives it offers, already in the introduction, not leaving the potential readers wondering if this review will be suited to their needs or not.
- In this respect, more info on types of papers reviewed and those not included, as well as the review timespan would be helpful.
- When considering the focus of this manuscript on steroidal hormones interacting with TRP channels, it is surprising that many important sources are not discussed here, for example:
https://doi.org/10.1038/ncb1208-1383
https://doi.org/10.1016/j.ygcen.2014.10.004
Clearly, more thorough referencing is still needed. - Many abbreviations are not defined anywhere in the manuscript, e.g., ICAM-1, COX-2, and PAI-1.
- On page 10: “to resolve the first 3D high structure” – Was this meant to discuss high-resolution structure? If so, I would suggest including also information on the actual resolution achieved (4Å) to highlight that it is not truly atomic-resolution as the high-resolution could imply to some readers.
- On page 22: “observed at the hypothalamus” I believe this should read “observed in the hypothalamus“
Author Response
Response to Reviewer 1 Comments
Point 1: The manuscript aims at reviewing the TRP-channels and their interactions with different steroid molecules, in a rather comprehensive manner. The structure of this review is appropriate, and it is supplemented with an adequate number of tables and figures.
My biggest concern is the novelty of this paper, in particular, given the broad interest in TRP channels, I dearly miss information on how the current review differs from hundreds of other reviews on this topic being published all the time. The authors should clearly define their manuscript, stating what new perspectives it offers, already in the introduction, not leaving the potential readers wondering if this review will be suited to their needs or not.
In this respect, more info on types of papers reviewed and those not included, as well as the review timespan would be helpful.
Response 1: We thank this reviewer for valuable comments that have strenghtned our manuscript. The types of studies reviewed and cited in our manuscript are strictly related to the regulation of TRP channels by steroids. We know that there are several reviews focused in TRP channels and lipids, the structure of TRP channels, etc. However, in the last 5 years only 3 reviews about TRP channels and steroids were published:
doi: 10.3389/fphar.2019.00419
doi: 10.3389/fmolb.2018.00073.
doi: 10.1016/j.ygcen.2014.10.004
One of them focuses on ion channels in general and steroids, the other is on the role of TRP channels and sex-related migrane pain and the last one is indeed about TRP channels and steroids in normal physiology and in disease (2015). Since the publishing of this 2015 review, there has been a considerable advance in the field, which we believe we have covered in our review.
Point 2: When considering the focus of this manuscript on steroidal hormones interacting with TRP channels, it is surprising that many important sources are not discussed here, for example:
https://doi.org/10.1038/ncb1208-1383
https://doi.org/10.1016/j.ygcen.2014.10.004
Response 2: Thank you for the comment. The references suggested are review articles, thus we decided it was better to cite the original experimental works.
For example, suggested reference with DOI:10.1038/ncb1208-1383, is a paper commenting on the original work performed in DOI:10.1038/ncb1801, which in our manuscript is reference 26. We have detailed this work in the section for the TRPM3 channel (lanes 564-572). Nonetheless, we have now added reference DOI:10.1038/ncb1208-1383 (reference #10 line 90), as suggested by the reviewer.
The source https://doi.org/10.1016/j.ygcen.2014.10.004 is one of the three reviews published in the last five years about TRP channels and steroids. Our manuscript is an update on this topic. We added the suggested reference as reference#11 (line 90), where we also highlighted the relevance of our manuscript and also added the following references:
doi:10.3389/fmolb.2018.00073 (reference #12 line 90).
doi:10.3390/molecules181012012 (reference #13 line 90).
Point 3: Clearly, more thorough referencing is still needed.
Response 3: Thank you for the observation. However, we think that we did an extensive review about the original/experimental works related to TRP channels and steroids, we think the most relevant information has been included in our manuscript to the best of our ability. The only studies that we had not cited were the reviews that you kindly suggested and other three reviews about TRP channels and steroids. However, we have now included them all (line 90).
Point 4: Many abbreviations are not defined anywhere in the manuscript, e.g., ICAM-1, COX-2, and PAI-1.
Response 4: Now all abbreviations are defined in parenthesis when mentioned for the first time.
Point 5: On page 10: “to resolve the first 3D high structure” – Was this meant to discuss high-resolution structure? If so, I would suggest including also information on the actual resolution achieved (4Å) to highlight that it is not truly atomic-resolution as the high-resolution could imply to some readers.
Response 5: Thanks for the observation. This paragraph was not meant to discuss high resolution structure since the main objective of this manuscript is not related to that topic. The main focus of our manuscript is to discuss two important aspects that modulate these channels: the regulation of their expression at the mRNA and protein levels by steroidal compounds and the regulation of TRP channel function by putative interaction of some steroids with binding sites in these channels. Additionally, this review has associated the TRP channel regulation by steroids with a physiological effect.
Also, following your suggestion we have rewritten this paragraph and it now reads: This tool enabled the determination of the first structure determination of a member of the TRP family, the TRPV1 channel with a resolution of 3.27 Å [5]. This strategy also facilitated solving the structures of other members of this family of ion channels [56-58] (lines 210-212)
Finally, we have added a section with a brief description of predicted steroid-binding pockets localized in some structures of TRP channels (as suggested the Rw2) (section 5, lanes 938-1002) and a section 6 where we discuss steroids´s effects in the temperature threshold of activation of some TRP channels. (lanes 1004-1039).
Point 6 . On page 22: “observed at the hypothalamus” I believe this should read “observed in the On page 22: “observed at the hypothalamus” I believe this should read “observed in the hypothalamus“
Response 6: We have changed “at” to “in”, this correction appears now on page 15, line 488.

Reviewer 2 Report
The authors have written a review that catalogues various interactions between the TRP family of channels and steroids, after beginning with a brief review of the biosynthesis of steroids. The review is extensive and includes a range of topics.
On the whole the review is useful, tying together several related strands on steroid-TRP channel interactions. There is more in this manuscript than appears to be in reviews on relevant subjects in the literature at present. Given this, it is likely that there will be sufficient interest for the manuscript to be worth publishing.
This manuscript suggests a number of questions for further investigation that are not even mentioned, and I am not certain how far the literature goes. It is obviously not appropriate to criticize a manuscript for not doing something that the authors never set out to do. However, it would be useful to also review some of the physical properties, especially the extremely large temperature dependence of at least some of the channels; this is not particularly commented on in the manuscript; there are multiple places in which it is noted that TRP channels sense temperature, but the only way this is related to steroids is in the conclusion that steroids may regulate the expression of the channels. Does this sensing change when the TRP channel complexes with a steroid? A couple of cases are cited in which there are complexes, but without any structural discussion. There are many thousands of TRP structures in the protein data bank. Structure is not discussed at all in the manuscript, except in general terms; the X-ray and cryo EM structures are not discussed. If there is nothing in the literature concerning these structures with relevance to steroid complex formation, that might have been worth a mention in itself; if the main steroid-TRP channel interaction concerns steroid complexing with receptors that effect gene transcription, some detail concerning the receptors would be interesting, albeit not as directly relevant. It would be possible to continue suggesting extensions, but these can be left for later work. However, it might have been worth mentioning whether there is any literature on these matters. The conclusions could include suggestions for further work.
The English is good, and the manuscript is clearly written. However, there are occasionally non-standard usages, such as unclassical for non-classical. There are a fair number of typos, and occasional missing words; even though the reader can infer the words from context, the manuscript could use a good proofreading.
The manuscript can be published with minor revisions.
Author Response
Response to Reviewer 2 Comments
Point 1: The authors have written a review that catalogues various interactions between the TRP family of channels and steroids, after beginning with a brief review of the biosynthesis of steroids. The review is extensive and includes a range of topics.
On the whole the review is useful, tying together several related strands on steroid-TRP channel interactions. There is more in this manuscript than appears to be in reviews on relevant subjects in the literature at present. Given this, it is likely that there will be sufficient interest for the manuscript to be worth publishing.
This manuscript suggests a number of questions for further investigation that are not even mentioned, and I am not certain how far the literature goes. It is obviously not appropriate to criticize a manuscript for not doing something that the authors never set out to do. However, it would be useful to also review some of the physical properties, especially the extremely large temperature dependence of at least some of the channels; this is not particularly commented on in the manuscript; there are multiple places in which it is noted that TRP channels sense temperature, but the only way this is related to steroids is in the conclusion that steroids may regulate the expression of the channels. Does this sensing change when the TRP channel complexes with a steroid?
Response 1. We thank this Reviewer for useful comments that we have tried to incorporate to the best of ability. We have now added a brief discussion about the complexes between TRP channels and steroids and their implication in temperature sensing (section 6, lines 1004-1039). It appears on page 27.
Point 2: A couple of cases are cited in which there are complexes, but without any structural discussion. There are many thousands of TRP structures in the protein data bank. Structure is not discussed at all in the manuscript, except in general terms; the X-ray and cryo EM structures are not discussed.
Response 2. We appreciate the observation but it is not within the scope of this manuscript to discuss high resolution structures of TRP channels, especially because there are several very nice reviews on this topic. In our review, we set out to discuss the discoveries that show the importance of regulation of the expression of these channels at the mRNA and protein levels by steroidal compounds and also the possible role of steroids as ligands of these channels. We now discuss some structural aspects of the interactions of some steroids and TRP channels at the end of the manuscript (section 5, lines 938-1002).
Point 3: If there is nothing in the literature concerning these structures with relevance to steroid complex formation, that might have been worth a mention in itself; if the main steroid-TRP channel interaction concerns steroid complexing with receptors that effect gene transcription, some detail concerning the receptors would be interesting, albeit not as directly relevant. It would be possible to continue suggesting extensions, but these can be left for later work. However, it might have been worth mentioning whether there is any literature on these matters. The conclusions could include suggestions for further work.
Response 3. Thanks for your observation. The most representative case where a steroid is a ligand of a TRP channel is that testosterone and TRPM8. This work has functional and biochemical evidence about the existence of a putative-steroid binding site (lines 760-764). The case for TRPA1 and androstenedione also provides partial evidence about the existence of a steroid binding site (lines 930-935). Although the experimental data suggest that PregS directly actives TRPM3, and this activation is stereoselective when the steroid is extracellularly applied, there is not a high-resolution structure of this channel nor any supporting information about a putative steroid-binding site. Similarly, there is little experimental evidence about the direct actions of PregS and DHEA in inhibiting TRPV1 activity.
Despite this, we have performed some molecular docking simulations in order to visualize where these steroids could bind in the cryo-EM reported. This allowed us to add some extra structural information about the interaction of steroids and TRP channels (lanes 938-1002). We focused on the interactions between TRPM8 and testosterone and TRPV1 and TRPA1 with androstenedione. Unfortunately, a TRPM5 high-resolution structure has not been resolved, thus we were unable to do the above described procedures for this channel. We have also added a figure to show docking simulations between steroids and the channels (Figure 8).
Point 4: The English is good, and the manuscript is clearly written. However, there are occasionally non-standard usages, such as unclassical for non-classical. There are a fair number of typos, and occasional missing words; even though the reader can infer the words from context, the manuscript could use a good proofreading.
Response 4. Thank you for the comment, we have tried to amend all the non-standard usages, all the “unclassical” words were substituted by “non-classical”. We have corrected all other writing issues as best possible.
The manuscript can be published with minor revisions.

Round 2
Reviewer 1 Report
The revised version properly address all my comments. I could only spot some minor language errors introduced into the revised parts.
- line 210: This tool enabled the determination of the first structure determination of a member of the TRP family >> This tool enabled the determination of the first structure of a member of the TRP family
- line 1015: duplicated reference [124]. [124].